# Decoding living systems: Reassessing crop model frontiers via biological dynamics and optimized phenotype

Edgar S. Correa [1,2,3]*

1 Pontificia Universidad Javeriana, School of Engineering, Bogota, Colombia, 2 UMR AGAP Institut, Univ. Montpellier, CIRAD, INRAE, Institut Agro, Montpellier, France, 3 CIRAD, UMR AGAP Institut, Montpellier, France

* e_correa@javeriana.edu.co

## Abstract

Modeling and optimizing phenotypic performance of biological systems demands understanding how physiological processes mediate genotype-by-environment interactions. While AI-driven approaches achieve predictive accuracy, they often function as black boxes that obscure biological causality. Process-based models address this limitation through explicit mechanistic representation, enabling both quantitative optimization and biological interpretation. This study contributes an inverse engineering framework with three integrated layers: sensitivity analysis validating biological coherence, genetic algorithm exploring virtual phenotypes to identify adaptive strategies, and similarity analysis quantifying routes from computational optima to field-validated cultivars. Sensitivity analysis identified eight genetic-based coefficients governing yield with robust rankings (95% CI width = 0.04). The genetic algorithm explored 5,364 virtual cultivars across 40 generations, revealing two strategies: extended growth (116 days) achieving 4,837 kg/ha under higher water availability (815 mm, field capacity 0.30), and shortened cycles (100–103 days) maintaining high efficiency (HI: 0.55–0.58) under water deficit (540 mm, field capacity 0.23)—covering 89% of the cultivation area. Similarity analysis against 21 field-validated cultivars identified WAB56−50 (70.7%) and DKAP2 (67.2%) as breeding candidates, quantifying a 22–30% genetic gap between current germplasm and computational optima. The framework, built upon 3 years of field characterization, compressed the evaluation and selection cycle, enabling adaptation across regional precipitation gradients identified through GMM-based classification. The principles demonstrated here extend across biological scales—from organismal phenotyping to cellular systems where biological dynamics can be modeled and traits measured.

**Data availability statement:** All relevant data are within the paper and its Supporting information files. Additional simulation outputs, environmental input files, and all code for replication are publicly available from the Zenodo repository (https://doi.org/10.5281/zenodo.18094655). The GitHub repository contains organized code to replicate all figures and tables: https://github.com/EdgarStevenC/Crop-Growth-Modelling. This study used publicly available datasets: soil data from SoilGrids (https://soilgrids.org), climate data from NASA POWER (https://power.larc.nasa.gov), and crop observations from Gérardeaux et al. (2021), who obtained all necessary permits for the original data collection.

**Funding:** This research was funded by the Agropolis Fondation through the CropModAdapt project (Contract No. 2201-026, 2023–2024), and by the ClimBeR initiative – France–CGIAR Action Plan on Climate Change (ICARDA Agreement No. 200303, 2023–2024). The funders had no role in study design, data analysis, decision to publish, or preparation of the manuscript.

**Competing interests:** The author has read the journal's policy on competing interests and declares the following. The author declares no personal financial competing interests beyond the funding sources listed in the Funding section. For transparency, the author discloses prior professional and supervisory/mentoring relationships within the doctoral research context in which this work was developed. The individuals named did not contribute to the manuscript's study design, data analysis, interpretation, decision to publish, or writing, and therefore do not meet authorship criteria for this article; they are disclosed solely because these relationships could reasonably be perceived as relevant to the peer review and editorial process, including the identification of potential non-independent evaluators. María Camila Rebolledo has had prior professional relationships with the author within the doctoral research context, including scientific discussions on project needs and potential applications; may have academic/professional interests related to this research. Julian Ramirez-Villegas and Alexandre B. Heinemann have had close professional ties and collaborations within the relevant research network for this work; these relationships could reasonably be perceived as affecting independence in any evaluative role related to this work. Myriam

## Introduction

Predicting and optimizing phenotypic performance in living systems requires mechanistic and dynamic understanding of how biological processes mediate genotype-by-environment interactions. This challenge spans biological scales, from cellular stress responses to organism-level adaptation, and demands integrative frameworks combining process-based modeling with computational optimization.

The implications extend from crop resilience under climate change to cellular dynamics in disease progression. As global food security demands rise and precision medicine advances, deciphering these genotype-environment interactions has become essential across the life sciences.

Recent advances in artificial intelligence, image processing, pattern recognition, and high-throughput phenotyping have accelerated data acquisition and predictive modeling [1–8]. Yet while AI-driven approaches achieve predictive accuracy, they often function as black boxes that obscure biological causality. Process-based models address this limitation through explicit mechanistic representation, enabling both quantitative optimization and biological interpretation.

Agriculture exemplifies this challenge: breeders must rapidly phenotype large populations and accurately target the most promising progeny [9–11]. Yet accurately predicting crop performance under diverse and variable conditions remains difficult due to complex genotype-by-environment (G×E) interactions [12–14].

Biological process-based models offer a systematic approach to this challenge by encoding biological mechanisms rather than relying solely on correlative patterns. Mechanistic crop models (MCMs) exemplify this approach, providing a structured framework to simulate plant development as a dynamic system, integrating physiological processes such as photosynthesis, biomass allocation, phenology, and soil–plant–atmosphere interactions [15,16]. MCMs such as DSSAT, APSIM, STICS, and AquaCrop simulate crop responses under different environmental and management scenarios through five core modules: assimilation, water/nutrient balance, biomass partitioning, grain formation, and phenology (Fig 1) [17–19].

MCMs have proven effective for climate adaptation research [20,21]. While they do not capture gene dynamics or 3D architecture at molecular scales, their capacity to simulate functional trait responses—phenology, biomass partitioning, stress tolerance—provides the biological foundation for ideotype design.

Traditional plant breeding has relied on two empirical strategies. The first, defect correction, eliminates undesirable traits such as disease susceptibility or lodging through backcrossing—improving what exists without defining what is optimal. The second, yield selection, identifies high-performing variants through iterative field trials across environments—discovering superior genotypes without predicting why they perform. Both approaches require 10–15 years and substantial resources, yet operate without explicit performance targets: breeders know what they find, not what they seek [22].

C. M. Donald proposed a third approach: predictive ideotype design. Rather than selecting from existing variation or correcting defects, this strategy defines optimal phenotypic configurations a priori using biological process-based models, then

Adam and Julian Colorado are former mentors/ supervisors during earlier stages of the doctoral research. The author affirms sole authorship of this work. No other competing interests are declared. There are no patents, products in development or marketed products associated with this research to declare. This does not alter the author's adherence to PLOS ONE policies on sharing data and materials.

identifies or creates genotypes capable of achieving them [10,23,24]. The approach inverts traditional breeding logic—from "select the best available" to "define the target and engineer toward it."

Genetic algorithms enable systematic exploration of this trait space through simulation. While molecular approaches optimize at the genotype level, process-based modeling captures genotype-by-environment causality, enabling regional-scale optimization across climate and soil gradients identified through systematic GMM-based environmental classification [25–28].

This study demonstrates an inverse engineering approach to phenotype optimization, using rainfed rice as a model system. The framework integrates three analytical layers: sensitivity analysis identifying control parameters, genetic algorithm optimization exploring fitness landscapes, and similarity analysis mapping routes from computational optima to existing germplasm. Unlike previous approaches centered primarily on yield-driven morphological optimization, a dual-performance metric is introduced, integrating Harvest Index (HI) and Water Use Efficiency (WUE), to quantify grain conversion efficiency relative to water availability.

Global sensitivity analysis using the Morris method identifies the most influential genotype-based parameters. These are optimized via Genetic Algorithm across 5,364 virtual cultivars and four distinct environments, systematically selected to represent 89% of the regional cultivation area based on soil-climate gradients [25]. The optimized ideotype is then compared against 21 field-characterized rice cultivars, spanning indica, japonica, and hybrid groups, using multidimensional similarity metrics [29,30]. This analysis identifies cultivars closest to computational optima, quantifying the genetic gap that breeding programs must traverse.

The integration of process-based modeling, AI-driven optimization, and genotypic similarity analysis provides a foundation that extends beyond agriculture to biological systems from organismal to cellular scales where measurable traits determine functional performance under environmental constraints.

## Materials and methods

### Biological-based growth modeling setup

The CERES-Rice model represents genotypic variation through eleven crop coefficients that govern phenological development, carbon partitioning, and stress responses (Fig 2). Of these, eight parameters (P1, P5, P2O, P2R, G1, G2, G3, PHINT) were selected for optimization based on sensitivity analysis, while thermal stress thresholds (THOT, TCLDP, TCLDF) were excluded due to context-dependent sensitivity. These coefficients translate genetic information into physiological behavior, enabling in-silico exploration of genotype × environment interactions.

Plant growth progression is quantified through Growing Degree Days (GDD), representing the thermal time required for phenological advancement. This thermal accumulation framework—calculated as the daily difference between mean temperature and a base temperature below which growth ceases—provides the mechanistic basis for simulating developmental plasticity across environments [34–37].

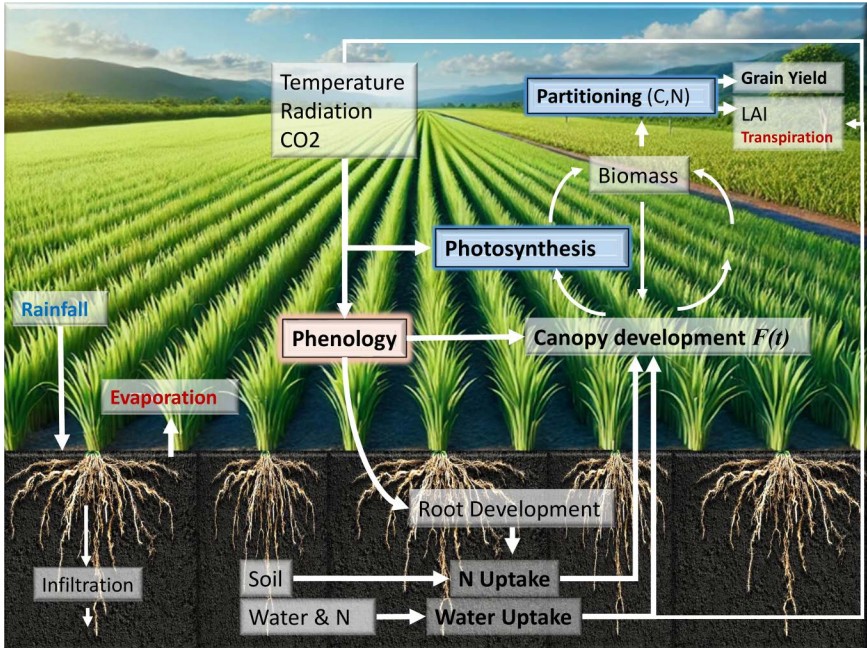

**Fig 1. Conceptual framework of biological process-based crop modeling: genotype-by-environment interactions unfold as environmental inputs drive physiological processes governed by genetic-based coefficients.** Source: Original diagram created by the author using AI-assisted tools.

The genetic coefficients are functionally organized into three groups reflecting their physiological roles:

**Phenological development parameters.** P1 (thermal time to panicle initiation, GDD) and P5 (grain filling duration, GDD) collectively determine crop cycle length from emergence to physiological maturity. P1 governs vegetative phase duration, directly influencing anthesis timing; extended vegetative periods promote greater biomass accumulation, enhanced stem reserves, and increased spikelet number. P2O (critical photoperiod, hours) and P2R (photoperiod-induced delay, GDD) modulate developmental rate in response to daylength [38,39].

**Source-sink partitioning parameters.** During vegetative growth, three parameters determine yield potential: PHINT (phyllochron interval, GDD) controls leaf appearance rate and canopy expansion [40,41]; G1 defines spikelet number per panicle; and G3 (tillering coefficient) governs productive tiller formation. During grain filling, G2 (potential grain weight, g) determines individual grain sink strength.

**Thermal stress parameters.** Spikelet sterility is modulated by temperature extremes through THOT (heat-induced sterility threshold), TCLDF (cold-induced sterility), and TCLDP (cold-induced panicle delay). Low temperatures (15–19°C) during panicle development impair pollen formation [42–45], while high temperatures (>35°C) during flowering disrupt fertilization [46–49]. These thresholds enable simulation of climate-induced yield reductions critical for adaptation assessment.

The strong interdependencies among parameters—where changes in vegetative duration (P1) cascade through tillering (G3), spikelet formation (G1), and grain filling dynamics (G2, P5)—create a high-dimensional optimization challenge that requires systematic computational exploration, motivating the sensitivity analysis and genetic algorithm framework described in the following sections.

## Model input data: Environment and cultivars

Process-based modeling captures genotype-by-environment interactions through a defined data architecture: environmental drivers as inputs, genetic coefficients as modulators, and phenotypic traits as outputs. This structure

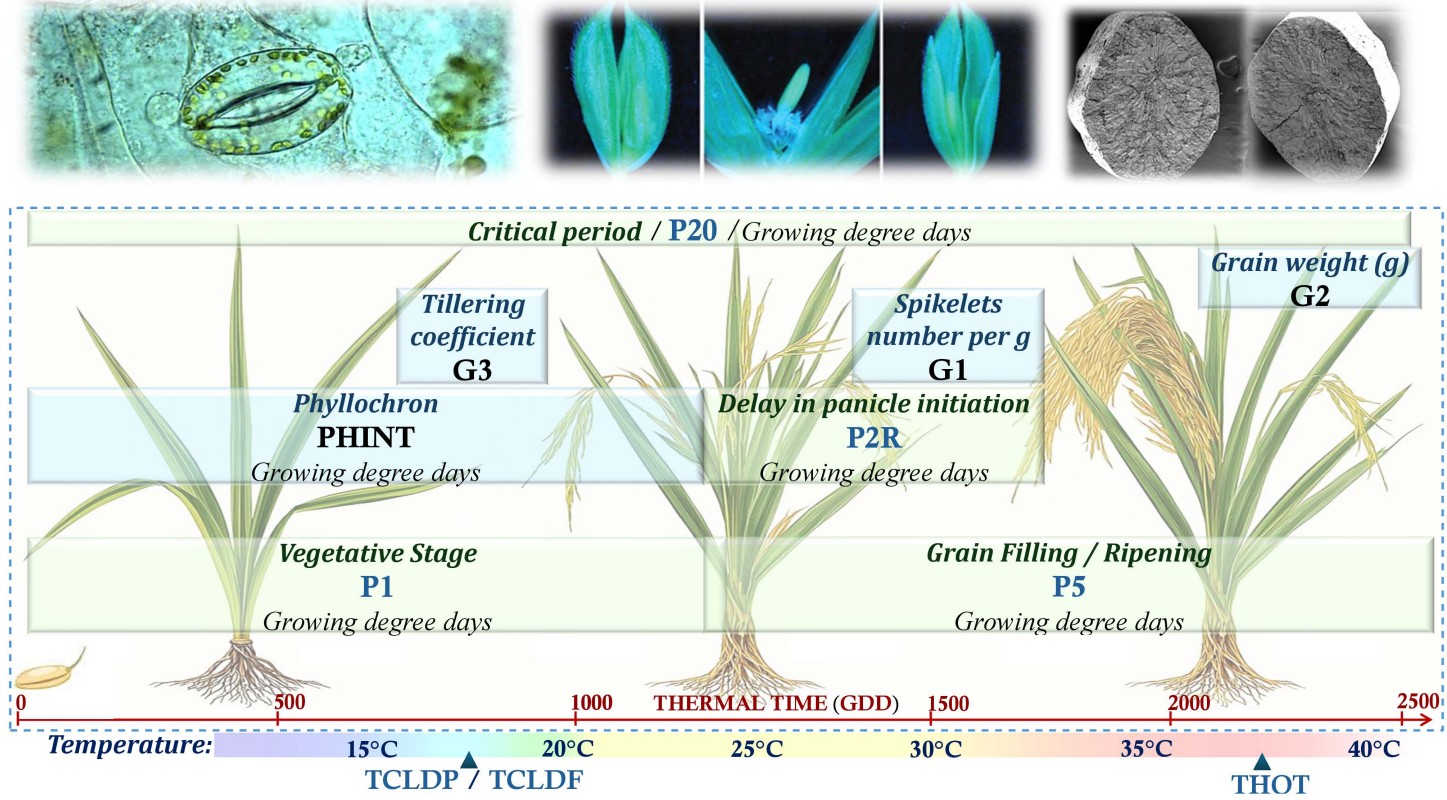

**Fig 2. Genetic coefficients in CERES-Rice organized by physiological function: phenological development (P1, P2O, P2R, P5), source-sink partitioning (PHINT, G1, G2, G3), and thermal stress thresholds (THOT, TCLDP, TCLDF).** Parameter interdependencies create a high-dimensional optimization landscape where sensitivity analysis guides parameter selection for genetic algorithm exploration. Source: Original diagram created by the author using AI-assisted tools. Microscopy images used for illustration are reprinted from Fan et al. (2023) [31], Liu et al. (2016) [32], and Yang et al. (2025) [33] under a CC BY license, with permission from the respective publishers, original copyright 2023, 2016, and 2025. All files are freely available online for use, distribution, and reproduction with proper attribution.

enables mechanistic simulation of how genotypes respond differentially to environmental conditions—a framework validated here through field experimentation where 21 cultivars were characterized across diverse soil-climate combinations.

This research is based on rainfed field experiments conducted between 2012 and 2014 along the west coast of Africa, covering the Casamance and Eastern Senegal, including Ziguinchor, Sédhiou, Kolda [50]. This study examines the crop's response to environmental factors by integrating soil properties and climatic variables with a comprehensive database that includes growth responses such as yield and biomass measurements across 21 varieties.

**Soil data:** Soil properties were analyzed through laboratory tests at a depth of 30 cm. In contrast, the satellite-based SoilGrids database provides soil characteristics at three depth levels (15 cm, 30 cm, and 100 cm), which are used as input variables for the crop growth model [51]. The properties include soil texture, essential nutrients, and other critical parameters, as outlined in Table 1. Additionally, the crop growth model requires soil hydraulic properties such as the permanent wilting point (SLLL), field capacity (SDUL), saturation (SSAT), and saturated hydraulic conductivity (SSKS), which are derived using Saxton and Rawls' pedotransfer functions [52–55].

**Climate data:** The crops and the mechanistic crop model representation (MCM) is driven by critical inputs such as minimum and maximum temperature (°C), solar radiation (MJ/m²/day), wind speed (m/s), relative humidity (%), and

**Table 1. Environmental inputs and phenotypic outputs for process-based modeling. Soil, climate, and crop parameters used in the crop growth model.**

| Environment | | Cultivar |
|---|---|---|
| **Soil Parameters** | **Climate Parameters** | **Crop Observations** |
| Clay content (g/kg) – SLCL | Minimum Temperature (°C) | Biomass (kg/ha) |
| Silt content (g/kg) – SLSI | Maximum Temperature (°C) | Grain Yield (kg/ha) |
| Soil Sand (g/kg) – Soil Sand | Solar Radiation (MJ/m²/day) | Number of Grains (#/m²) |
| Bulk density (cg/cm³) – SBDM | Wind Speed (m/s) | Number of Tillers (#/m²) |
| Coarse fraction (cm³/dm³) – SLCF | Relative Humidity (%) | Anthesis (days) |
| Soil Nitrogen (cg/kg) – SLNI | Precipitation (mm/day) | Maturity (days) |
| Total Organic Carbon (dg/kg) – SLOC | | |
| pH (unitless) – SLHW | | |

precipitation (mm/day), all of which are derived from the rainfed rice study [50]. For regional-scale applications, these meteorological variables are sourced from the NASA POWER dataset [56].

*Crop observations:* To study crop growth in response to the environment (GxE), it is crucial to understand the response of each genotype to environmental conditions. The rainfed rice study [50] analyzes 21 varieties and reports the crop's response in terms of variables such as organic matter accumulation including grain yield, biomass, number of grains, and number of tillers, as well as phenology variables like anthesis and maturity.

### GMM-based environmental classification

A study conducted in Casamance and Eastern Senegal identified twelve distinctive environments using Principal Component Analysis (PCA) and a Gaussian Mixture Model (GMM) in three dimensions, considering all soil and meteorological variables [25]. Fig 3 shows that 89% of the region is represented by four environments, covering two soil types and two climatic zones. The convex hull is the polygon that encloses all given points in an environment, and the closest point to the centroid is selected as the representative sample for that environment.

Soil type 1, predominantly located in the eastern part of the region, is characterized by higher pH (5.91), bulk density (1.50 g/cm³), silt content (30.53 g/kg), and clay content (26.33 g/kg). It also presents intermediate hydraulic properties related to water retention, with SLLL (0.16 cm³/cm³), SDUL (0.29 cm³/cm³), SSAT (0.44 cm³/cm³), and a moderate saturated hydraulic conductivity (SSKS, 0.75 cm/h). In contrast, soil type 2 is classified as sandy (56.59 g/kg), with higher saturated hydraulic conductivity (SSKS, 1.09 cm/h) and a slightly lower pH (5.87). Due to its soil composition, soil type 2 exhibits low water retention capacity, which facilitates rapid water infiltration and drainage when saturated.

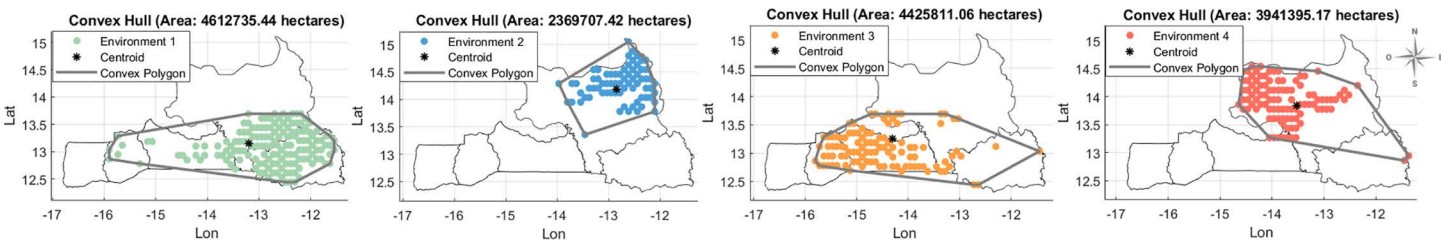

**Fig 3. Environmental categories in the Casamance and Eastern Senegal region.** Base map layers (coastlines and administrative boundaries) were derived from publicly available datasets distributed with MATLAB Mapping Toolbox and are compatible with CC BY 4.0 licensing. Environmental classification and data layers were adapted from Correa et al. (2025), published under a CC BY license [25]. Map generated using MATLAB R2024b with Mapping Toolbox and Image Processing Toolbox.

Climate Type 1 predominates in the southern region and is characterized by higher relative humidity (80.33%), lower average solar radiation (19.16 MJ/m²/day) with lower variability (3.44 MJ/m²/day), and moderate temperatures, including minimum (22.19°C), maximum (30.60°C), and average (26.39°C) values, along with greater accumulated precipitation (917 mm). Climate Type 2 is mainly located in the northern part of the study area and is distinguished by 44% lower accumulated rainfall (513.24 mm) and 18% lower average relative humidity (67.98%), along with higher average solar radiation (19.62 MJ/m²/day), and higher minimum (22.51°C), maximum (33.04°C), and average (27.78°C) temperatures compared to the southern climate. The vapor pressure deficit (VPD), calculated from temperature and humidity, was 77% higher in northern climates (1.19 kPa) compared to southern climates (0.67 kPa), indicating substantially greater atmospheric water demand that intensifies drought stress during critical reproductive stages.

## Software simulation setup

All simulations were performed on a workstation equipped with an Intel Core i7-12700H processor (12th Gen, 2.30 GHz) and 32 GB RAM, running Windows 11 Professional (Build 26100.7171). CERES-Rice simulations were executed using DSSAT v4.8 [57,58]. Genetic algorithm optimization and post-processing analyses were implemented in MATLAB R2024b [59] and R v4.4.2 [60].

## Sensitivity analysis of CERES-Rice model

Quantifying parameter influence on system performance is essential for process-based biological modeling, enabling identification of key drivers that govern phenotypic outcomes.

The Morris method was selected for its computational efficiency and ability to screen multiple parameters simultaneously [61,62]. Unlike local sensitivity approaches that perturb one parameter from a fixed baseline, the Morris method explores the entire parameter space through randomized trajectories, capturing potential interactions between physiological processes encoded in the model.

The method operates by generating $r$ trajectories through the 11-dimensional parameter space, where each trajectory consists of $k + 1$ points (Algorithm 1, Fig 4a). Consecutive points differ in exactly one parameter by a step size $\Delta$, allowing

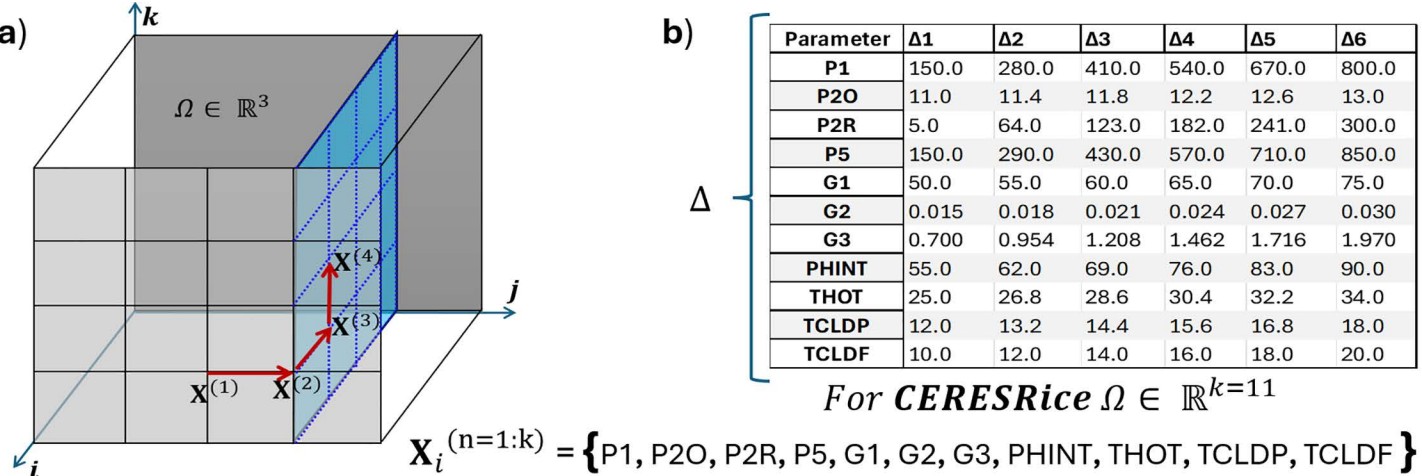

**Fig 4. Morris method sampling strategy for CERES-Rice sensitivity analysis.** a) Trajectory generation in parameter space (illustrated for dimensions); consecutive points differ by one parameter, enabling elementary effect computation. b) Step size () and bounds for each of the 11 physiological genetic-based coefficients, defined from ranges reported for rice cultivars.

computation of elementary effects—the change in model output attributable to each parameter perturbation. For the 11 CERES-Rice physiological coefficients, this design required $r(k + 1) = r \times 12$ model evaluations per trajectory, substantially fewer than factorial designs while maintaining comprehensive parameter coverage.

**Algorithm 1. Morris method for sensitivity analysis of CERES-Rice genetic coefficients**

```
1: Input: Genetic coefficients k = 11, trajectories r, step size Δ (Fig 4b)
2: Output: Elementary effects for each coefficient-output combination
3: Initialize: Random base point x* within coefficient bounds
4: for each trajectory t = 1 to r do
5:   Generate initial point x⁽¹⁾ from x*
6:   for i = 2 to k + 1 do
7:     Modify one coefficient by ±Δ to obtain x⁽ⁱ⁾
8:     Evaluate CERES-Rice model at x⁽ⁱ⁾
9:     Compute elementary effect for modified coefficient
10:   end for
11: end for
12: Return: Distribution of elementary effects per genetic coefficient
```

Parameter bounds and step sizes (Fig 4b) were defined based on physiological ranges reported for rice cultivars, ensuring that perturbations remained within biologically plausible limits. Each parameter was perturbed across its full range, from early-maturing to late-maturing phenotypes for phenological traits (P1, P5), and from low to high tillering capacity for reproductive traits (G3).

To ensure robust sensitivity estimates, 20 independent replications were performed with randomized parameter sequences, providing confidence intervals for parameter rankings (S1 Table).

The Relative Sensitivity Index (RSI) was computed to standardize sensitivity across output variables with different units and magnitudes. For each parameter-output combination, RSI quantifies the mean absolute change in output relative to the maximum observed change (Eq 1):

$$\text{RSI} = \frac{1}{n} \sum_{i=1}^{n} \frac{|\Delta Y_i|}{\max(|\Delta Y_i|)}$$

(1)

where $\Delta Y_i$ represents the change in model output (grain yield, biomass, anthesis, maturity) resulting from the $i$-th elementary effect. RSI values range from 0 (no influence) to 1 (maximum influence), enabling direct comparison of parameter importance across outputs with different units (kg/ha for yield, days for phenology).

This standardized metric enables identification of which physiological traits—developmental timing, canopy formation, or reproductive capacity—most strongly govern system performance, providing the foundation for targeted parameter optimization.

## Genetic algorithm optimization

The Genetic Algorithm (GA) is an artificial intelligence approach based on heuristic search for optimization problems. It is used with the CERES-Rice crop growth model to characterize the ideotype that optimizes grain conversion efficiency and water use efficiency by optimizing genetic crop growth parameters through the evolution of a population of candidate solutions over multiple generations for each environment. The algorithm starts by initializing a population of individuals, each representing a potential set of genetic parameters for the crop growth model. It evaluates the fitness of these individuals by assessing the harvest index (HI) and water use efficiency (WUE), which are integrated into the fitness function. The selection process favors the best-performing individuals, which are then used to generate the next generation through crossover and mutation operations. Mutation introduces diversity into the population, preventing premature convergence.

**Fitness Metric: HI-WUE Integrated Index.** To guide ideotype optimization and evaluate each set of genetic crop parameter combinations, an integrated efficiency index (HI-WUE) is employed as the fitness metric. This index combines two critical agronomic performance indicators: the Harvest Index (HI) and Water Use Efficiency (WUE), as defined in Eq 3. The HI quantifies the efficiency of assimilate partitioning by representing the ratio of grain yield to total biomass. In contrast, the WUE reflects the efficiency of water utilization, expressed as the amount of grain produced per unit of accumulated evapotranspiration. The HI-WUE index enables an integrated assessment of genotypic performance in terms of both yield potential and resource use efficiency.

To ensure equal contribution of both metrics to the fitness function, WUE was normalized to [0,1] scale using physiologically grounded bounds from the literature:

$$WUE_{norm} = \frac{WUE - WUE_{min}}{WUE_{max} - WUE_{min}} \tag{2}$$

Where $WUE_{min}$ and $WUE_{max}$ represent physiological bounds (2 and 15 kg ha$^{-1}$ mm$^{-1}$, respectively) reported for aerobic rice under drought adaptation [63]. The integrated fitness function is then:

$$HI\text{-}WUE = HI + WUE_{norm} \tag{3}$$

This formulation ensures balanced optimization with equal weighting (50%–50%) between grain conversion efficiency and water use efficiency.

## Algorithm 2. Genetic Algorithm for ideotyping in CERES-Rice

```
1: Initialize population P with Num_ind individuals
2: Set global best ind_best and min_dist_global to infinity
3: for generation=1 to Num_P do
4:   Evaluate fitness of each individual in P
5:   for each individual i in P do
6:     if fitness of individual i is better than current best then
7:       Update ind_best min_dist_global
8:     end if
9:   end for
10:  P ← Select_parents(P)
11:  P ← Crossover_parents(P)
12:  P ← Mutation(P, Thr_mut)
13:  Track and plot dist_hist and dist_hist_mean
14: end for
15: Save ind_best and plot final fitness results
```

The Genetic Algorithm for ideotyping in the CERES-Rice crop model is outlined in Algorithm 2. It iterates over multiple generations, optimizing genetic-based crop parameters to enhance resource efficiency and yield performance across different environments. Throughout this process, the algorithm refines the population by selecting parents, applying crossover, and introducing mutations to evolve toward the optimal ideotype. Key performance metrics, including maximum fitness, mean population fitness, and the best genotype's fitness, are continuously tracked to assess progress and convergence.

The algorithm uses the following parameters: the number of generations, set to 40; the initial population size, set to 15; and the mutation probability, set to 0.7. These parameters ensured adequate exploration of the 8-dimensional parameter search space (P1, P5, P2R, PHINT, P2O, G1, G2, G3). The high mutation rate facilitated broad search space coverage, while convergence analysis (S1 Fig) demonstrates fitness stabilization by generations 5–13 across environments, confirming successful identification of optimal parameter combinations.

Each individual in the GA represents a potential solution, corresponding to a specific combination of genetic crop parameters for the CERES-Rice crop model. The individual is defined as $X_i = \{P1, P5, P2R, PHINT, P2O, G1, G2, G3\}$, where each parameter takes a value within a predefined range, as shown in Fig 4-b.

The GA employs roulette wheel selection for parent selection, arithmetic crossover for recombination, and adaptive mutation targeting high-sensitivity parameters identified through sensitivity analysis. Detailed mathematical formulations and implementation of these operators are provided in Supplementary Methods.

## Similarity analysis with field-characterized cultivars

Fig 5 presents 21 rice cultivars from a rainfed experiment conducted in the region [50], classified into three genetic groups: *indica*, *japonica*, and *hybrid*. Each cultivar is characterized by eight genetic coefficients (P1, P5, P2R, PHINT, P2O, G1, G2, G3). The optimized ideotypes were compared against these cultivars to identify genotypic similarities and inform breeding recommendations.

Similarity between ideotypes and field-characterized cultivars was assessed using three complementary distance metrics: Euclidean distance measuring direct separation in parameter space, Manhattan distance summing absolute differences along each dimension, and Cosine distance assessing angular similarity between coefficient vectors. These metrics capture different aspects of genetic proximity—geometric distance, coordinate-wise deviation, and directional alignment—providing robust cultivar ranking through metric consensus.

The similarity index for each ideotype-cultivar pair was calculated as the normalized average across all distance metrics (Eq 4). Distance values were inverted and scaled to yield similarity scores in the interval [0, 1], where higher values indicate greater genetic proximity.

$$\text{Similarity} = 1 - \frac{\bar{d}_{ij}}{\max(\bar{d})} \tag{4}$$

where $\bar{d}_{ij}$ represents the average distance between ideotype $i$ and cultivar $j$ across all metrics, and $\max(\bar{d})$ is the maximum average distance observed.

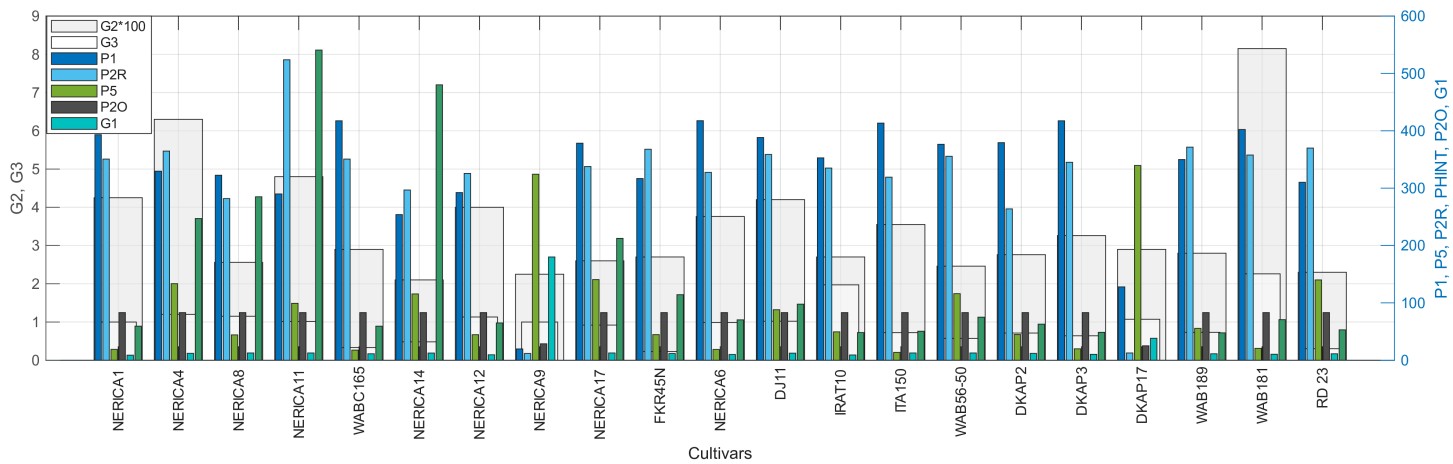

**Fig 5. Genetic crop growth coefficients for 21 rice cultivars characterized through rainfed field experiments.** Cultivars are classified by genetic group: *indica* (green), *japonica* (blue), and hybrids (orange). These field-validated parameters serve as the reference panel for similarity analysis against computationally optimized ideotypes.

Frequency analysis quantified how often each cultivar appeared among the top-ranked matches across all ideotypes and metrics, identifying cultivars with consistent proximity to multiple optimized phenotypes. Principal Component Analysis (PCA) was applied to the combined dataset of ideotypes and cultivars to visualize genetic distances in reduced dimensional space, with Euclidean distance in PCA space providing an independent validation of similarity rankings.

Algorithm 3 presents the structured approach for identifying the most genetically similar cultivars. The similarity index and frequency serve as complementary metrics: similarity quantifies genetic proximity, while frequency indicates relevance across the ideotype population.

## Algorithm 3. Identification of the most similar cultivars

```
1: Input: Virtual cultivar dataset V, real cultivar dataset R, distance metrics D
2: Output: Frequency analysis and similarity index visualization
         ▷Step 1: Compute distance and find closest cultivars
3: for each metric d ∈ D do
4:   for each virtual cultivar v ∈ V do
5:     Compute distances: dist[v, r] ← d(v, r),   ∀r ∈ R
6:     Sort R by dist[v, r] in ascending order
7:     Select the top-k closest cultivars and store in T[v, d]
8:   end for
9: end for
         ▷Step 2: Aggregate closest cultivars for frequency analysis
10: for each virtual cultivar v ∈ V do
11:   Concatenate T[v, :] into a single list of cultivars
12:   Compute frequency of each cultivar
13:   Sort cultivars by frequency in descending order
14: end for
         ▷Step 3: Compute normalized similarity index
15: for each virtual cultivar v ∈ V do
16:   for each real cultivar r ∈ T[v, :] do
17:     Compute similarity: similarity[v, r] = 1 − (avg_distance[v,r] / max(avg_distance))
18:   end for
19:   Normalize similarity scores relative to the highest value
20: end for
21: End Algorithm
```

## Results

This section presents the key findings of the optimization of the CERES-Rice crop model. Sensitivity analysis quantifies the impact of genetic-based crop parameters on biomass accumulation, grain yield, grain number, tiller count, and critical phenological stages, including anthesis and maturity. The ideotyping optimization process determines the optimal combination of genetic parameters, while similarity analysis, based on field-characterized cultivars, provides targeted recommendations for selecting the most promising progeny and potential genetic modifications to enhance resilience across diverse environments in the region.

### Sensitivity analysis of genetic parameters

Sensitivity analysis, conducted across 20 independent replications with randomized parameter sequences, quantified the influence of 11 genetic coefficients on six model outputs (Fig 6). Parameter rankings were robust (mean 95% CI width = 0.04; S1 Table).

**Biomass and yield sensitivity.** **Biomass accumulation** was primarily governed by parameters controlling canopy development and vegetative phase duration. PHINT (phyllochron interval) exhibited the highest sensitivity (RSI = 0.70 ± 0.02), reflecting its direct regulation of leaf appearance rate and consequent light interception capacity. P1 (vegetative

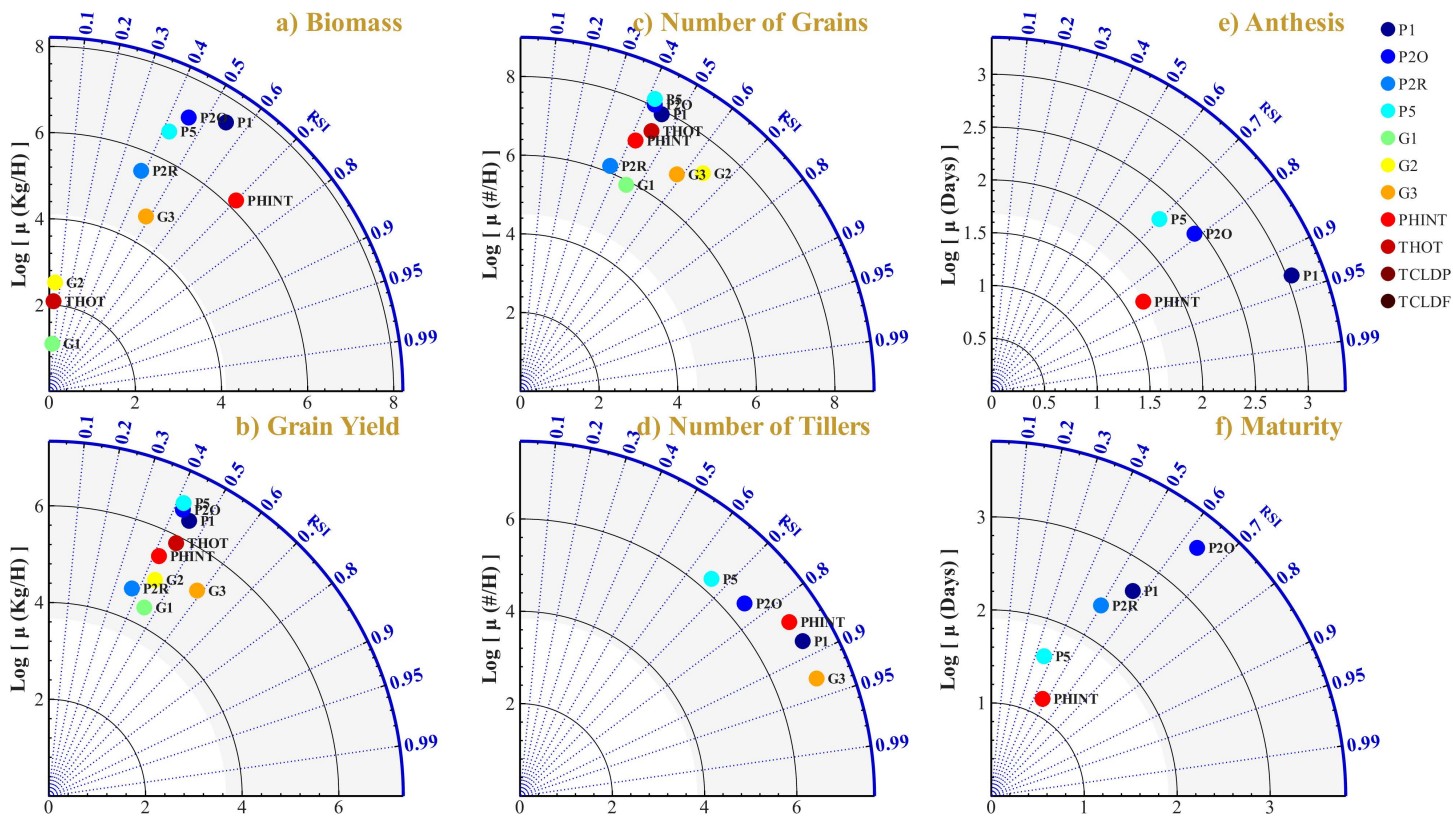

**Fig 6. Sensitivity analysis reveals hierarchical parameter control over crop performance.** Phenological parameters (P1, PHINT) dominate developmental timing and biomass accumulation, while reproductive parameters (G3, G2) govern yield component formation. Thermal stress parameters (TCLDP, TCLDF) exhibit negligible sensitivity under the studied conditions. Relative Sensitivity Index (RSI) for CERES-Rice model outputs: a) Biomass, b) Grain Yield, c) Number of Grains, d) Number of Tillers, e) Anthesis, f) Maturity. Values represent means across 20 independent replications; complete statistics with 95% CI in S1 Table.

thermal time) ranked second (RSI = 0.55 ± 0.04), determining the duration of the photosynthetically active phase. G3 (tillering coefficient) demonstrated moderate influence (RSI = 0.48 ± 0.05) through its effect on tiller-derived leaf area contribution. P2O and P5 contributed moderately (RSI = 0.45 and 0.42, respectively). Grain-related parameters G1 and G2 exhibited negligible sensitivity (RSI < 0.07), indicating that sink traits do not exert feedback regulation on source accumulation within the CERES-Rice model structure.

**Grain yield** exhibited a more distributed sensitivity pattern, reflecting its dependence on both source and sink processes. G3 demonstrated the highest sensitivity (RSI = 0.59 ± 0.04), as tiller number directly determines panicle density and yield potential. P1 ranked second (RSI = 0.46 ± 0.04), influencing yield through vegetative biomass accumulation and spikelet differentiation. THOT (heat-induced sterility threshold) exhibited substantial sensitivity (RSI = 0.45 ± 0.06), indicating potential yield reduction under elevated temperature conditions during flowering. Number of grains was most sensitive to G2 (potential grain weight, RSI = 0.64 ± 0.03) and G3 (RSI = 0.59 ± 0.04), confirming the predominance of sink-related parameters. Number of tillers was predominantly controlled by G3 (RSI = 0.93 ± 0.01), followed by P1 (RSI = 0.88 ± 0.02) and PHINT (RSI = 0.84 ± 0.03), demonstrating strong genetic determination of tillering capacity.

**Phenological timing. Anthesis timing** was governed by phenological parameters with minimal estimation uncertainty. P1 exhibited the highest sensitivity across all output-parameter combinations (RSI = 0.93 ± 0.02), followed by PHINT

(RSI = 0.86 ± 0.02) and P2O (RSI = 0.79 ± 0.12). Reproductive parameters (G1, G2, G3) exhibited zero sensitivity for anthesis, consistent with the model structure wherein flowering date is independent of sink-related traits.

**Maturity timing** displayed a broader sensitivity distribution, with P2O ranking highest (RSI = 0.64 ± 0.08) followed by P1 (RSI = 0.57 ± 0.19). The elevated variability observed for P1 sensitivity on maturity (CI width = 0.38) compared to anthesis (CI width = 0.03) reflects uncertainty accumulation through the grain-filling phase.

**Thermal stress parameters.** Cold stress parameters TCLDP and TCLDF exhibited near-zero sensitivity across all outputs (RSI ≈ 0.00), indicating that cold-induced spikelet sterility was not a limiting factor under the thermal conditions characteristic of Senegal (minimum temperatures 22–23°C during the reproductive phase). THOT demonstrated moderate sensitivity exclusively for grain yield and grain number (RSI ≈ 0.45), suggesting localized susceptibility to heat stress during anthesis that merits consideration under projected climate change scenarios.

The narrow confidence intervals (mean CI width = 0.04) across 20 replications confirm robust parameter rankings, validating the selection of P1, P2O, P2R, P5, G1, G2, G3, PHINT as optimization targets for the genetic algorithm.

## Ideotype optimization through genetic algorithm

The genetic algorithm explored a phenotypic landscape of 5,364 virtual cultivars across 40 generations, systematically identifying optimal genetic coefficient combinations that maximize the HI-WUE index—a composite metric integrating harvest index and water use efficiency. This computational exploration systematically evaluated unique genotype-environment combinations, demonstrating the potential of process-based modeling coupled with AI optimization.

Optimization was conducted across four distinct environments identified through GMM-based environmental classification [25], collectively representing 89% of the Casamance and Eastern Senegal cultivation area. These environments capture the critical climate-soil gradients constraining rainfed rice production: southern humid zones (Env 1, 3: precipitation ~815 mm) versus northern drought-prone regions (Env 2, 4: precipitation ~540 mm), crossed with soil water retention capacity ranging from high (Env 1: SDUL = 0.30 cm³/cm³) to low (Env 4: SDUL = 0.23 cm³/cm³).

**Efficiency-yield correlations.** Fig 7 illustrates the relationship between the HI-WUE index and agronomic performance across all evaluated genotypes. Correlation analysis revealed a clear functional hierarchy: traits directly incorporated in the fitness function—harvest index ($R^2$ = 0.88–0.97) and water use efficiency ($R^2$ = 0.86–0.93)—exhibited strong positive correlation with HI-WUE. Grain yield demonstrated consistently strong correlation ($R^2$ = 0.78–0.86, $p < 0.001$), confirming that optimizing physiological efficiency translates to productivity gains. Number of grains showed moderate, environment-dependent correlation ($R^2$ = 0.39–0.60), reflecting sink contribution to yield formation.

In contrast, biomass ($R^2 < 0.04$), leaf area index ($R^2 < 0.05$), root density ($R^2 < 0.02$), and phenological timing ($R^2 < 0.08$) showed negligible correlation with HI-WUE within the optimization. This pattern is consistent with sensitivity analysis findings where reproductive parameters (G1, G2, G3) controlled yield components independently of vegetative biomass accumulation (RSI < 0.07). Complete correlation statistics are provided in S2 Table (Supplementary Material).

**Environment-specific convergence dynamics.** Convergence analysis revealed optimization dynamics shaped by environmental constraints (S1 Fig, Supplementary Material). Southern environments with higher precipitation showed contrasting patterns: Env 1 (highest water retention: SDUL = 0.30 cm³/cm³) required 23 generations to reach 95% of maximum fitness, while Env 3 (sandy soil with high organic carbon) converged fastest at generation 10, achieving the highest fitness value (HI-WUE = 0.973).

Northern environments under drought stress (Env 2: 557 mm; Env 4: 525 mm precipitation) both converged at generation 20 with similar fitness values (0.955 and 0.948, respectively). These environments yielded identical optimal genetic coefficients despite differing soil properties (SDUL: 0.28 vs 0.23 cm³/cm³).

**Two distinct adaptive strategies.** Fig 8 presents the predicted performance of optimized ideotypes. Two distinct strategies emerged from the optimization:

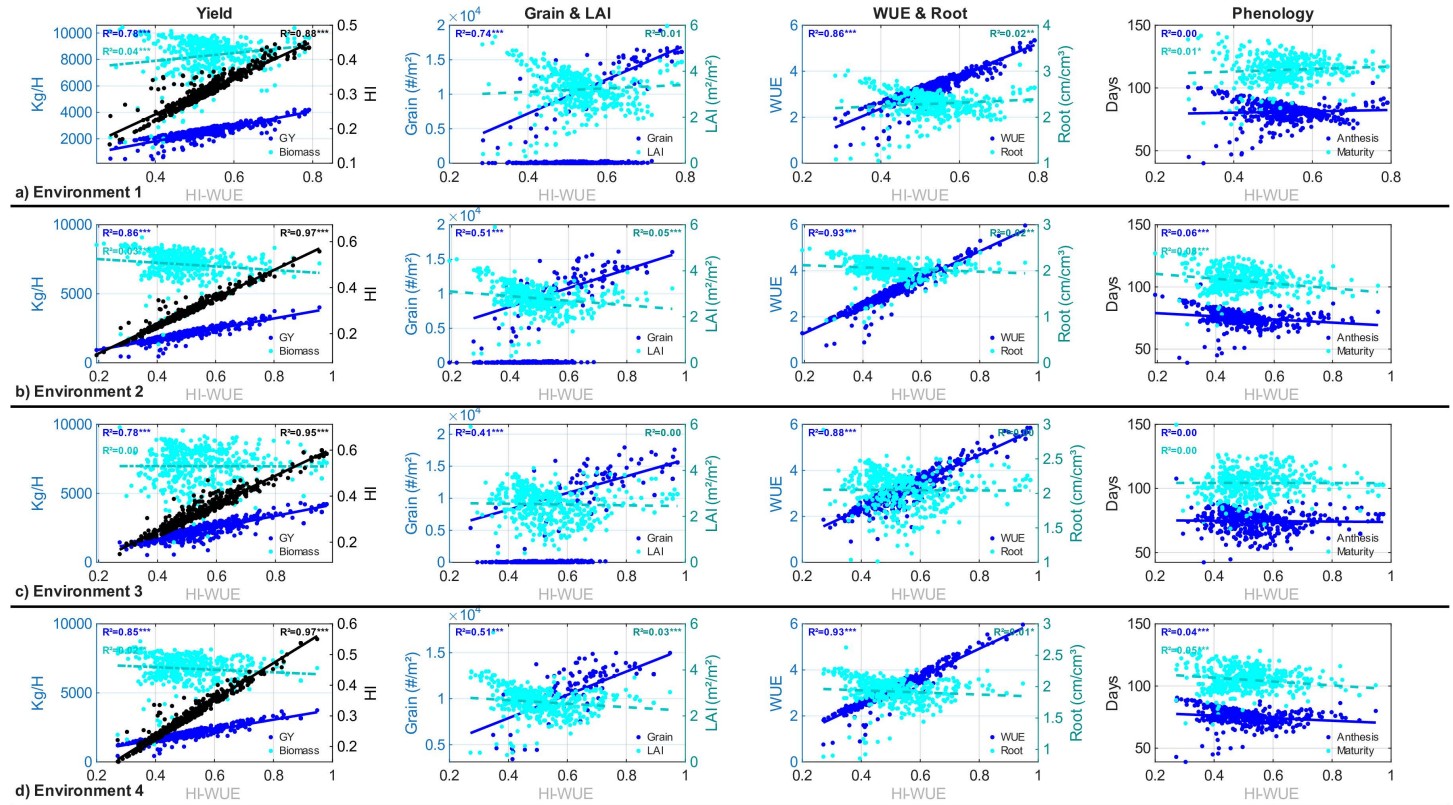

**Fig 7. Phenotypic landscape explored by genetic algorithm optimization across four contrasting environments.** Each point represents a virtual cultivar evaluated through CERES-Rice simulation (n = 5,364). The HI-WUE index demonstrates strong correlation with grain yield (R² = 0.78–0.86, p 0.001), harvest index (R² = 0.88–0.97, p 0.001), and water use efficiency (R² = 0.86–0.93, p 0.001). Biomass, root architecture, and phenological timing exhibit negligible correlation (R² 0.08), revealing that efficiency optimization operates through assimilate partitioning rather than source accumulation. Complete correlation statistics in S2 Table (Supplementary Material).

*Strategy A—Extended growth (Env 1, 30% of cultivation area):* Under favorable conditions combining high soil water retention with adequate precipitation, the optimal ideotype featured extended grain filling duration (P5 = 372 GDD) and moderate phyllochron (PHINT = 74 GDD). This strategy achieved the highest yield (4,837 kg/ha) with HI = 0.54 and WUE = 6.17 $kg\ ha^{-1} mm^{-1}$ across a 116-day cycle.

*Strategy B—Shortened cycle (Env 2–4, 59% of cultivation area):* Under water-limited conditions, the algorithm converged toward reduced grain filling duration (P5 = 207–248 GDD) and similar phyllochron (PHINT = 72–74 GDD), completing the reproductive cycle in 100–103 days. These ideotypes achieved yields of 3,743–4,213 kg/ha with harvest index of 0.55–0.58 and water use efficiency of 5.84–5.97 $kg\ ha^{-1} mm^{-1}$.

**Reproductive coefficient stability.** Across all environments, reproductive coefficients remained stable (G1 ≈ 62, G2 ≈ 0.025 g, G3 ≈ 0.90) while phenological parameters varied with environmental conditions (S3 Table, Supplementary Material).

## Similarity and affinity of optimized ideotypes

Each variety and ideotype is characterized by a set of genetically-driven crop growth parameters: *P*1, *P*5, *P*2R, *PHINT*, *P*2O, *G*1, *G*2, *G*3. Fig 9-a presents a 3D visualization of the Principal Component Analysis (PCA) conducted on these genetic parameters, which accounts for 85.8% of the total variance. The four optimized ideotypes

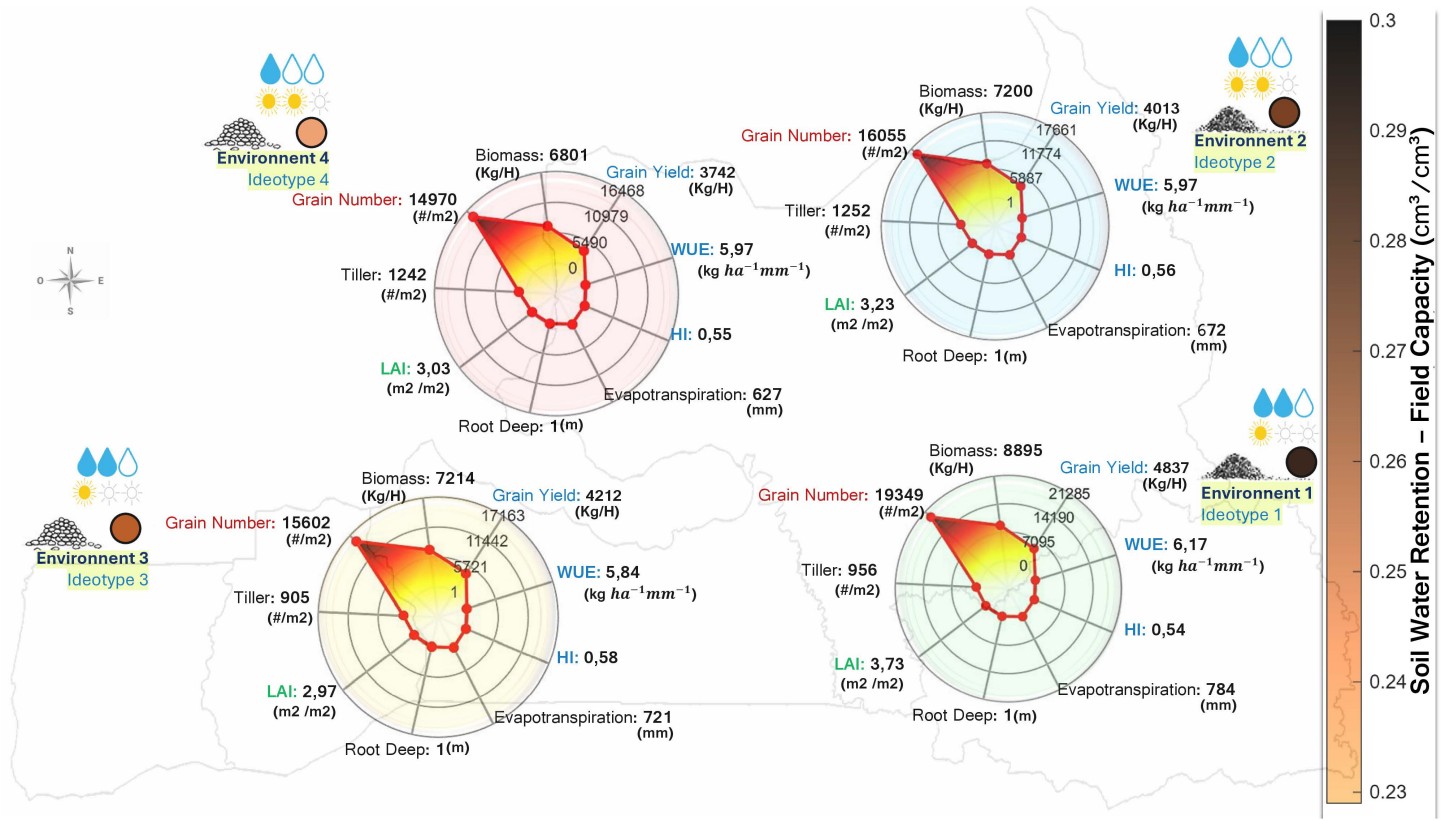

**Fig 8. Optimized ideotype performance across four environments characterized by the intersection of soil water retention (high to low) and precipitation regime (815 mm in the south vs. 540 mm in the north).** The optimization identified two distinct adaptive strategies: extended growth exploiting favorable conditions (Env 1) and drought-escape phenology under water limitation (Env 2–4). Reproductive parameters remained stable across environments while phenological coefficients required environment-specific tuning—consistent with the hierarchical parameter control identified through sensitivity analysis. Complete genetic coefficients are reported in S3 Table (Supplementary Material). Base map layers (coastlines and administrative boundaries) were derived from publicly available datasets distributed with MATLAB Mapping Toolbox and are compatible with CC BY 4.0 licensing. Environmental classification and data layers were adapted from Correa et al. (2025), published under a CC BY license [25]. Map generated using MATLAB R2024b with Mapping Toolbox and Image Processing Toolbox.

(ID1–ID4), representing the highest-performing genetic configurations for each environment, are shown in red. Field-validated cultivars are classified into three genetic groups: *indica* (green), *japonica* (blue), and hybrids (orange). Dotted lines connect each ideotype to its four nearest cultivars based on Euclidean distance in PCA space. Fig 9-b depicts the similarity scores and frequencies of these varieties across all distance metrics, illustrating the alignment of genetic traits.

A single variety, **WAB56−50**, achieved the highest global similarity (70.7%) and appeared consistently across all four ideotypes, making it the most promising candidate for breeding programs. **DKAP2** ranked second with 67.2% average similarity and was identified as the nearest cultivar for Ideotypes 2, 3, and 4 based on PCA distance.

For Ideotype 1 (favorable environment), the top matches were WAB56−50 (77.9%), RD 23 (74.0%), and NERICA17 (73.5%). Ideotypes 2 and 4 (drought-dominated environments) converged to identical genetic configurations, with DKAP2 (72.8%), WABC165 (67.4%), and WAB56−50 (66.9%) as the closest cultivars. Ideotype 3 showed intermediate characteristics with WAB56−50 (71.1%), NERICA17 (68.9%), and DKAP2 (66.0%) as top matches.

The complete similarity matrix between all ideotypes and field-validated cultivars is presented in Fig 10. Red boxes highlight the top five cultivars with highest average similarity across all ideotypes. Three varieties appeared in three or

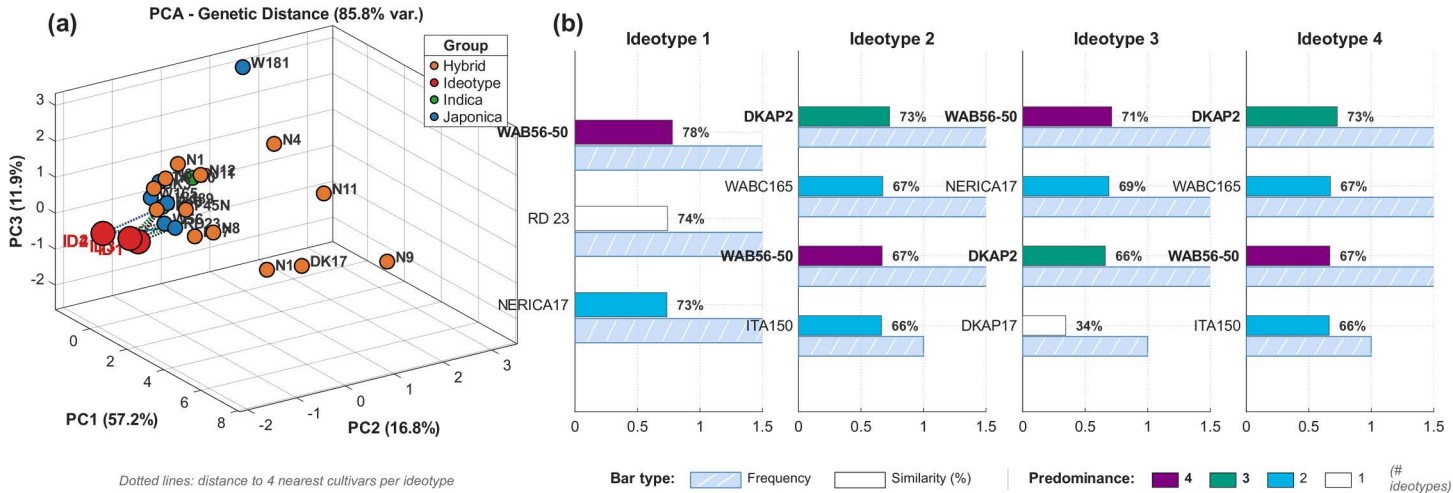

**Fig 9. Genetic similarity between optimized ideotypes and field-validated cultivars. (a)** PCA of eight genetic coefficients (85.8% variance explained). Dotted lines connect ideotypes (ID1–ID4) to their nearest cultivars. Colors indicate genetic groups: *indica* (green), *japonica* (blue), hybrids (orange), and ideotypes (red). **(b)** Top cultivar matches ranked by similarity (solid bars) and frequency across metrics (striped bars). Bar color shows predominance across ideotypes (4: purple to 1: white). WAB56−50 and DKAP2 are the most promising breeding candidates.

more ideotype rankings: **WAB56−50** (all 4 ideotypes), **DKAP2** (3 ideotypes), and **RD 23** (2 ideotypes). The convergence of Ideotypes 2 and 4 suggests that drought stress is the dominant selective pressure in these environments, leading to similar optimal genetic configurations centered around shortened reproductive phases and enhanced water use efficiency.

Fig 11 presents the parameter-by-parameter comparison between optimized ideotypes and the five highest-affinity cultivars. The 22–30% genetic gap separating current cultivars from computational optima is not uniformly distributed: phenological parameters (P1, P5, P2R) show the largest divergence, while reproductive coefficients (G1, G2, G3) are already near-optimal. This hierarchy defines breeding priorities—sink traits can be maintained through marker-assisted selection while phenological adaptation requires targeted crossing with environment-specific donors.

## Discussion

Three findings structure this discussion: the biological validation provided by sensitivity analysis, the adaptive strategies revealed by optimization, and the breeding roadmap defined by similarity analysis. Each analytical layer informed the next, progressively translating computational predictions into actionable recommendations.

### Sensitivity analysis as biological validation

The hierarchical sensitivity patterns observed—phenological parameters governing development, reproductive parameters controlling yield components, and thermal parameters showing context-dependent influence—serve as internal validation of the model's physiological coherence. A process-based model that failed to reproduce these expected functional separations would indicate structural deficiencies in its biological representation. This interpretability distinguishes process-based approaches from purely statistical or black-box methods, where predictor influence can be quantified but its biological meaning depends on external interpretation rather than explicit model structure.

The near-zero sensitivity of cold stress parameters (TCLDP, TCLDF) under Senegalese conditions, contrasted with moderate heat stress sensitivity (THOT), demonstrates that sensitivity analysis identifies which biological processes are active versus inactive in specific environmental contexts. This diagnostic capacity extends beyond parameter ranking to inform model applicability across environmental gradients—critical for process-based modeling of living systems under

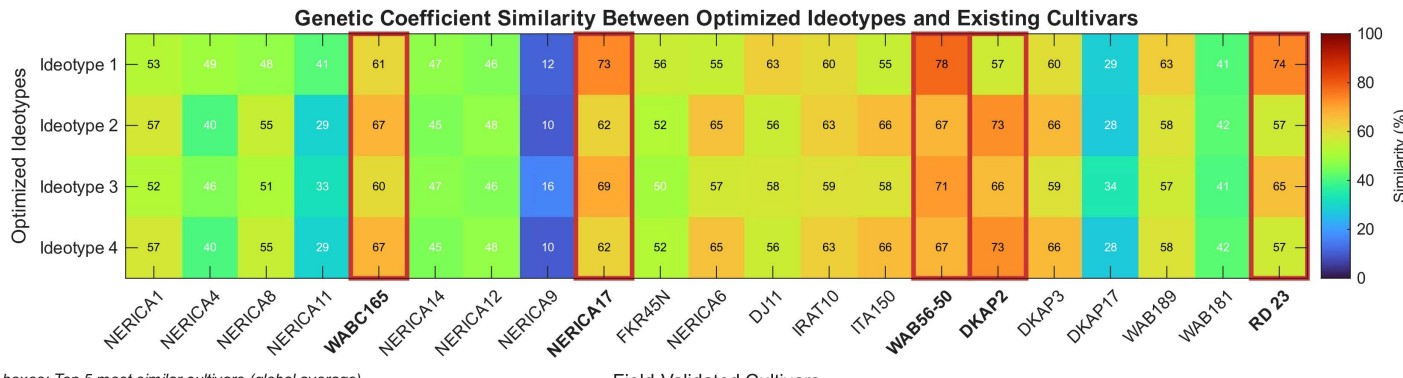

**Fig 10. Similarity heatmap between optimized ideotypes and field-validated cultivars.** Each cell represents the average similarity (%) computed from Euclidean, Manhattan, and Cosine distance metrics. Values above 50% are displayed in black text, below 50% in white. Red boxes indicate the top five cultivars with highest global similarity: WAB56−50 (70.7%), DKAP2 (67.2%), NERICA17 (66.6%), WABC165 (64.1%), and RD 23 (63.4%). Bold cultivar names denote high-affinity candidates.

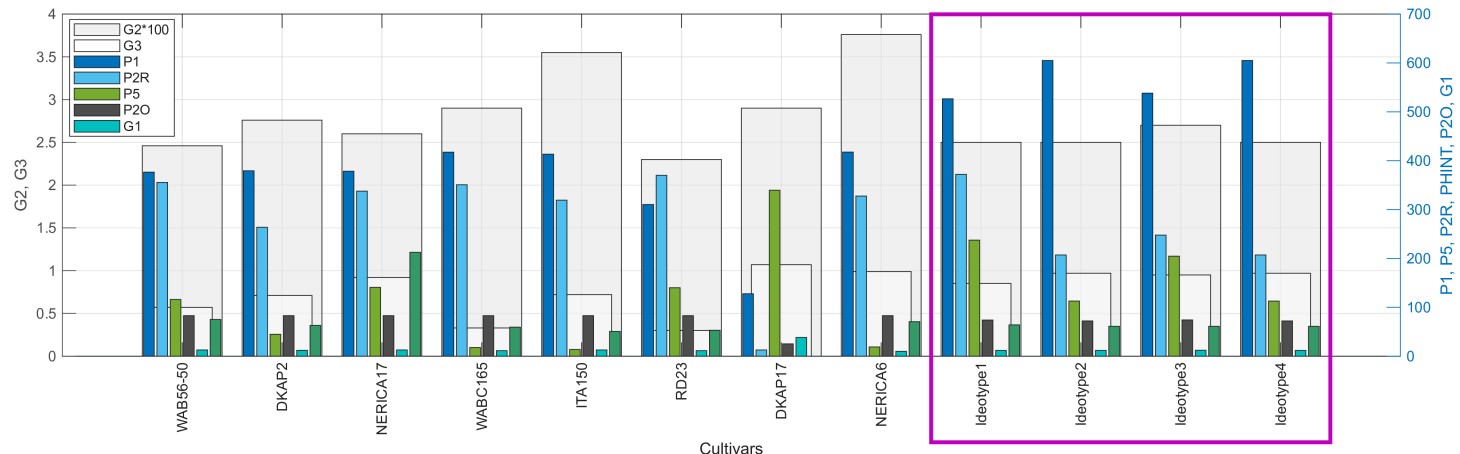

**Fig 11. Genetic crop growth parameters of optimized ideotypes and highest-affinity cultivars.** Comparison of the eight genetic coefficients (P1, P5, P2R, PHINT, P2O, G1, G2, G3) between the four optimized ideotypes (ID1–ID4) and the top field-validated cultivars identified through similarity analysis: WAB56-50, DKAP2, NERICA17, RD 23, and WABC165. Parameter values are normalized for visual comparison.

variable conditions. The narrow confidence intervals achieved through 20 replications (mean CI width = 0.04) confirm that sensitivity rankings are robust features of the model structure rather than random fluctuations, ensuring reliable parameter selection for subsequent optimization.

## From parameter influence to phenotypic optimization

The sensitivity results directly informed the optimization strategy: the eight parameters exhibiting highest sensitivity indices (P1, P5, P2R, P2O, G1, G2, G3, PHINT) were selected for genetic algorithm exploration, while thermal stress parameters were excluded based on their negligible influence under Senegalese conditions. This principled parameter selection reduced search space dimensionality while preserving the degrees of freedom most influential for ideotype performance.

The genetic algorithm evaluated 5,364 virtual cultivars across 40 generations. Strong correlations between the HI-WUE fitness index and grain yield ($R^2 = 0.78$–0.86, $p < 0.001$) validate the framework's capacity to translate physiological

efficiency into productivity gains. Two distinct adaptive strategies emerged: extended growth cycles (P5 = 372 GDD, 116 days) achieving maximum yields (4,837 kg/ha) in favorable southern environments, and shortened drought-escape cycles (P5 = 207–248 GDD, 100–103 days) maintaining high efficiency (HI: 0.55–0.58) in water-limited northern regions.

The climatic contrast between regions extends beyond precipitation totals. Vapor pressure deficit (VPD)—the driving force for transpiration—reaches 1.19 kPa in northern environments compared to 0.67 kPa in the south, representing 77% greater atmospheric water demand (Supplementary Note: VPD Calculation). This elevated VPD accelerates soil water depletion during the growing season, explaining why phenological escape rather than extended growth optimizes performance under northern conditions.

The convergence of Environments 2 and 4 to identical genetic coefficients despite differing soil properties (SDUL: 0.28 vs 0.23 cm³/cm³) reveals that precipitation deficit dominates over soil water-holding capacity in determining optimal phenological strategy. This finding simplifies breeding program targeting: a single drought-adapted cultivar can serve 59% of the Casamance and Eastern Senegal cultivation area. The stability of reproductive coefficients (G1 ≈ 62, G2 ≈ 0.025 g, G3 ≈ 0.90) across all environments, while phenological parameters varied substantially, provides a mechanistic basis for breeding: sink traits can be selected universally, phenological adaptation requires environment-specific tuning.

### Bridging computational ideotypes to breeding targets

Optimization identifies optimal phenotypes; translating these into breeding targets requires mapping the genetic distance between computational ideotypes and existing germplasm. The 70.7% maximum similarity achieved by WAB56-50 indicates that even the closest cultivar requires traversing 29.3% of the genetic parameter space to reach ideotypic configuration. This quantified gap, not a limitation but a roadmap, defines the breeding challenge with unprecedented precision.

The distinct clustering of optimized ideotypes in PCA space (Fig 9a), explaining 85.8% of total variance (PC1: 57.2%, PC2: 16.8%, PC3: 11.9%), reveals that optimal genetic configurations are not represented in current germplasm. This finding carries dual implications: existing cultivars, despite decades of empirical selection, have not converged toward computationally identified optima; and substantial genetic gains remain accessible through targeted parameter modification. The consistency between PCA distance rankings (WAB56-50: d = 0.79 to ID1; DKAP2: d = 1.08–1.28 to ID2–4) and similarity scores computed from multiple metrics (Euclidean, Manhattan, Cosine) provides cross-validation that identified breeding candidates are robust to methodological choice.

WAB56-50 and DKAP2 emerge as complementary breeding candidates: WAB56−50 (*japonica*) aligns with the extended-growth strategy for favorable environments (Ideotype 1, 30% of cultivation area), while DKAP2 (hybrid) matches the drought-escape phenology required for water-limited zones (Ideotypes 2–4, 59% of area). The parameter-by-parameter comparison reveals a clear hierarchy for breeding intervention: phenological parameters require major adjustment (P1: + 29%, P2R: + 56%, P5: –21%), while reproductive parameters are already near-optimal (G1: + 4%, G3: –1%, P2O: + 3%). This decoupling—first identified in sensitivity analysis, confirmed in optimization convergence, and now quantified against existing germplasm—enables accelerated breeding: sink traits maintained through marker-assisted selection, phenological adaptation achieved through targeted crossing with environment-specific donors.

### Model boundaries and future directions

The framework's predictive capacity operates within CERES-Rice boundaries. The model captures dominant agronomic relationships but simplifies dynamic source-sink feedback, root architectural plasticity, and leaf-level physiological processes. The negligible correlation between biomass and HI-WUE (R² < 0.04) suggests that optimized ideotypes achieve efficiency through improved assimilate partitioning rather than enhanced source capacity—a pattern that must be interpreted within these structural constraints. Future developments integrating these mechanisms could refine ideotype predictions under conditions where belowground traits and source-sink signaling determine yield stability.

The similarity analysis is constrained by the 21-cultivar reference panel. Expanding this panel to include additional African germplasm—particularly drought-tolerant Sahelian landraces—could identify cultivars with higher similarity to water-limited ideotypes. Furthermore, linking genetic coefficients to quantitative trait loci (QTL) would enable marker-assisted selection toward ideotypic configurations, bridging the phenotypic parameters used here with genomic breeding tools.

The 22–30% genetic gap between existing cultivars and optimized ideotypes quantifies both opportunity and challenge. Conventional breeding requires 10–15 years to traverse such distances through recurrent selection. The computational framework demonstrated here—sensitivity analysis informing optimization, optimization identifying targets, similarity analysis mapping routes—compresses the discovery phase, allowing breeding resources to focus on the irreducible biological timescales of crossing, selection, and field validation. This acceleration, demonstrated here for drought-stressed rice, extends to any biological system where physiological processes can be modeled and traits measured.

## Conclusion

### AI-driven optimization of biological processes

This study demonstrates that AI-driven optimization, coupled with process-based modeling, can transform how we approach complex biological systems. The framework compressed a discovery phase, evaluating 5,364 virtual phenotypes across 40 generations. This acceleration does not replace biological understanding—it amplifies it. The genetic algorithm did not operate as a black box; it explored a fitness landscape defined by explicit physiological mechanisms, ensuring that identified optima are biologically interpretable and experimentally tractable.

The integration of sensitivity analysis, optimization, and similarity analysis creates a pipeline where each computational layer informs the next: mechanistic understanding guides parameter selection, optimization reveals adaptive strategies, and similarity analysis maps implementation routes through existing biological material. This architecture—pattern recognition constrained by mechanistic structure—represents a paradigm for applying AI to living systems without sacrificing interpretability.

### Case study findings

Applied to drought-stressed rice across four contrasting environments in Senegal, the framework revealed three actionable insights:

(i) Phenological parameters (P1, P5, P2R, PHINT) govern environmental adaptation and require local calibration, while reproductive parameters (G1, G2, G3) remain stable across conditions—enabling a two-track breeding strategy of universal sink selection with environment-specific developmental tuning.

(ii) Precipitation deficit overrides soil water-holding capacity as the dominant selective pressure. Ideotypes 2 and 4 converged to identical genetic coefficients despite differing soil properties, demonstrating that a single drought-adapted cultivar can serve 59% of the regional cultivation area.

(iii) WAB56-50 (70.7% similarity) and DKAP2 (67.2%) emerged as complementary breeding candidates, with a quantified 22–30% genetic gap defining the selection pressure required to achieve computational optima—transforming vague improvement goals into measurable breeding targets.

### Modeling living systems: Limitations and horizons

Living systems do not follow simple rules. They adapt, compensate, and exhibit emergent behaviors that challenge simplified modeling approaches. CERES-Rice captures dominant agronomic relationships but simplifies source-sink feedback,

root plasticity, and molecular stress responses. These simplifications constrain prediction where belowground traits and cellular signaling determine outcomes—precisely where the frontier lies.

Platforms like DSSAT—integrating models such as CERES-Rice—have served as frameworks for decades, enabling agronomic research worldwide. Yet widespread adoption has paradoxically constrained scientific progress: users apply these models as established tools rather than testable hypotheses, accepting default parameterizations without questioning underlying assumptions. Biological expertise without computational scrutiny accepts model outputs uncritically; engineering expertise without biological insight optimizes parameters blindly. The frontier lies where both converge—yet this intersection remains largely unexplored. Model boundaries stay unexamined, gaps unquantified, limitations unaddressed.

Cellular systems offer the next horizon. Proliferation rates, metabolic fluxes, stress responses—measured with increasing precision through advances in microscopy, high-throughput phenotyping, and single-cell technologies—generate data streams that process-based models have yet to integrate. The convergence of observation technologies with AI-driven pattern recognition creates unprecedented potential: data revealing what equations cannot capture and mechanistic frameworks ensuring biological coherence.

The principles demonstrated here are scale-independent. What changes across biological scales is not the analytical logic but the resolution of observation. Bridging this gap—from crop canopy to cellular architecture, from field phenotyping to subcellular dynamics—represents both the challenge and the opportunity for quantitative biology driven by technological advances and AI in the coming decade.

## Supporting information

**S1 File. Supplementary Material.** Supporting figure and additional details (includes S1 Fig with panels A–D: genetic algorithm convergence across four environments).
(PDF)

**S1 Table. Relative Sensitivity Index (RSI) from 20 Morris method replications.** Bold indicates the highest sensitivity per output.
(PDF)

**S2 Table. Pearson correlation ($R^2$) between the HI–WUE index and phenotypic outputs across four environments.** Statistical significance: *** $p<0.001$, ** $p<0.01$, * $p<0.05$, ns $p\geq0.05$. Correlations with $R^2<0.10$ are considered practically negligible despite statistical significance due to large sample sizes ($n>1,300$ per environment).
(PDF)

**S3 Table. Optimal genetic coefficients and predicted crop performance for each environment.** Env1: high water retention, high precipitation (30% area); Env2: medium retention, low precipitation (18%); Env3: medium retention, high precipitation (21%); Env4: low retention, low precipitation (20%). Values represent the best individual identified across 40 generations of genetic algorithm optimization.
(PDF)

## Acknowledgments

The author expresses deep gratitude to the School of Engineering at Pontificia Universidad Javeriana for providing a full Ph.D. scholarship and to the UMR-AGAP Institute (Genetic Improvement and Adaptation of Mediterranean and Tropical Plants) for institutional support and access to research facilities. The author also thanks CIRAD for providing the Ph.D. scholarship that significantly contributed to this research. Special recognition is given to Michael Dingkuhn for his valuable guidance and generosity in sharing expertise in crop modelling and scientific communication. The author also thanks Edward Gerardeaux for providing crop parameters and observational data, as detailed in Gerardeaux et al. (2021), and

Loyola Rodríguez Perez for helpful feedback and insightful discussions in plant physiology during meetings at the Plant Physiology Laboratory (Pontificia Universidad Javeriana, Bogotá).

## Author contributions

**Conceptualization:** Edgar S. Correa.

**Data curation:** Edgar S. Correa.

**Formal analysis:** Edgar S. Correa.

**Investigation:** Edgar S. Correa.

**Methodology:** Edgar S. Correa.

**Project administration:** Edgar S. Correa.

**Resources:** Edgar S. Correa.

**Software:** Edgar S. Correa.

**Validation:** Edgar S. Correa.

**Visualization:** Edgar S. Correa.

**Writing – original draft:** Edgar S. Correa.

**Writing – review & editing:** Edgar S. Correa.

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
