## [Decision Letter · Decision Letter 0]

8 Sep 2025

Dear Dr. Correa,

Thank you for submitting your manuscript to PLOS ONE. After careful consideration, we feel that it has merit but does not fully meet PLOS ONE’s publication criteria as it currently stands. Therefore, we invite you to submit a revised version of the manuscript that addresses the points raised during the review process.

We look forward to receiving your revised manuscript.

Kind regards,

Paulo Eduardo Teodoro, Dr.

Academic Editor

PLOS ONE

**Journal Requirements:**

1. When submitting your revision, we need you to address these additional requirements. Please ensure that your manuscript meets PLOS ONE's style requirements, including those for file naming. The PLOS ONE style templates can be found at https://journals.plos.org/plosone/s/file?id=wjVg/PLOSOne_formatting_sample_main_body.pdf and https://journals.plos.org/plosone/s/file?id=ba62/PLOSOne_formatting_sample_title_authors_affiliations.pdf 2. In your Methods section, please provide additional information regarding the permits you obtained for the work. Please ensure you have included the full name of the authority that approved the field site access and, if no permits were required, a brief statement explaining why. 3. Please update your submission to use the PLOS LaTeX template. The template and more information on our requirements for LaTeX submissions can be found at http://journals.plos.org/plosone/s/latex. 4. We note that the grant information you provided in the ‘Funding Information’ and ‘Financial Disclosure’ sections do not match. When you resubmit, please ensure that you provide the correct grant numbers for the awards you received for your study in the ‘Funding Information’ section. 5. Please know it is PLOS ONE policy for corresponding authors to declare, on behalf of all authors, all potential competing interests for the purposes of transparency. PLOS defines a competing interest as anything that interferes with, or could reasonably be perceived as interfering with, the full and objective presentation, peer review, editorial decision-making, or publication of research or non-research articles submitted to one of the journals. Competing interests can be financial or non-financial, professional, or personal. Competing interests can arise in relationship to an organization or another person. Please follow this link to our website for more details on competing interests: http://journals.plos.org/plosone/s/competing-interests 6. Thank you for uploading your study's underlying data set. Unfortunately, the repository you have noted in your Data Availability statement does not qualify as an acceptable data repository according to PLOS's standards. At this time, please upload the minimal data set necessary to replicate your study's findings to a stable, public repository (such as figshare or Dryad) and provide us with the relevant URLs, DOIs, or accession numbers that may be used to access these data. For a list of recommended repositories and additional information on PLOS standards for data deposition, please see https://journals.plos.org/plosone/s/recommended-repositories. 7. We note that Figure 1 in your submission contain copyrighted images. All PLOS content is published under the Creative Commons Attribution License (CC BY 4.0), which means that the manuscript, images, and Supporting Information files will be freely available online, and any third party is permitted to access, download, copy, distribute, and use these materials in any way, even commercially, with proper attribution. For more information, see our copyright guidelines: http://journals.plos.org/plosone/s/licenses-and-copyright. We require you to either present written permission from the copyright holder to publish these figures specifically under the CC BY 4.0 license, or remove the figures from your submission: a. You may seek permission from the original copyright holder of Figure 1 to publish the content specifically under the CC BY 4.0 license. We recommend that you contact the original copyright holder with the Content Permission Form (http://journals.plos.org/plosone/s/file?id=7c09/content-permission-form.pdf) and the following text:“I request permission for the open-access journal PLOS ONE to publish XXX under the Creative Commons Attribution License (CCAL) CC BY 4.0 (http://creativecommons.org/licenses/by/4.0/). Please be aware that this license allows unrestricted use and distribution, even commercially, by third parties. Please reply and provide explicit written permission to publish XXX under a CC BY license and complete the attached form.” Please upload the completed Content Permission Form or other proof of granted permissions as an "Other" file with your submission. In the figure caption of the copyrighted figure, please include the following text: “Reprinted from [ref] under a CC BY license, with permission from [name of publisher], original copyright [original copyright year].” b. If you are unable to obtain permission from the original copyright holder to publish these figures under the CC BY 4.0 license or if the copyright holder’s requirements are incompatible with the CC BY 4.0 license, please either i) remove the figure or ii) supply a replacement figure that complies with the CC BY 4.0 license. Please check copyright information on all replacement figures and update the figure caption with source information. If applicable, please specify in the figure caption text when a figure is similar but not identical to the original image and is therefore for illustrative purposes only. 8. We note that Figure 2 in your submission contain map images which may be copyrighted. All PLOS content is published under the Creative Commons Attribution License (CC BY 4.0), which means that the manuscript, images, and Supporting Information files will be freely available online, and any third party is permitted to access, download, copy, distribute, and use these materials in any way, even commercially, with proper attribution. For these reasons, we cannot publish previously copyrighted maps or satellite images created using proprietary data, such as Google software (Google Maps, Street View, and Earth). For more information, see our copyright guidelines: http://journals.plos.org/plosone/s/licenses-and-copyright. We require you to either present written permission from the copyright holder to publish these figures specifically under the CC BY 4.0 license, or remove the figures from your submission: a. You may seek permission from the original copyright holder of Figure 2 to publish the content specifically under the CC BY 4.0 license. We recommend that you contact the original copyright holder with the Content Permission Form (http://journals.plos.org/plosone/s/file?id=7c09/content-permission-form.pdf) and the following text:“I request permission for the open-access journal PLOS ONE to publish XXX under the Creative Commons Attribution License (CCAL) CC BY 4.0 (http://creativecommons.org/licenses/by/4.0/). Please be aware that this license allows unrestricted use and distribution, even commercially, by third parties. Please reply and provide explicit written permission to publish XXX under a CC BY license and complete the attached form.” Please upload the completed Content Permission Form or other proof of granted permissions as an "Other" file with your submission. In the figure caption of the copyrighted figure, please include the following text: “Reprinted from [ref] under a CC BY license, with permission from [name of publisher], original copyright [original copyright year].” b. If you are unable to obtain permission from the original copyright holder to publish these figures under the CC BY 4.0 license or if the copyright holder’s requirements are incompatible with the CC BY 4.0 license, please either i) remove the figure or ii) supply a replacement figure that complies with the CC BY 4.0 license. Please check copyright information on all replacement figures and update the figure caption with source information. If applicable, please specify in the figure caption text when a figure is similar but not identical to the original image and is therefore for illustrative purposes only.The following resources for replacing copyrighted map figures may be helpful: USGS National Map Viewer (public domain): http://viewer.nationalmap.gov/viewer/The Gateway to Astronaut Photography of Earth (public domain): http://eol.jsc.nasa.gov/sseop/clickmap/Maps at the CIA (public domain): https://www.cia.gov/library/publications/the-world-factbook/index.html and https://www.cia.gov/library/publications/cia-maps-publications/index.html NASA Earth Observatory (public domain): http://earthobservatory.nasa.gov/ Landsat: http://landsat.visibleearth.nasa.gov/ USGS EROS (Earth Resources Observatory and Science (EROS) Center) (public domain): http://eros.usgs.gov/# Natural Earth (public domain): http://www.naturalearthdata.com/ 9. If the reviewer comments include a recommendation to cite specific previously published works, please review and evaluate these publications to determine whether they are relevant and should be cited. There is no requirement to cite these works unless the editor has indicated otherwise. ?

Reviewers' comments:

**Comments to the Author**

1. Is the manuscript technically sound, and do the data support the conclusions?

Reviewer #1: Yes

Reviewer #2: Yes

Reviewer #3: Partly

2. Has the statistical analysis been performed appropriately and rigorously?

Reviewer #1: Yes

Reviewer #2: Yes

Reviewer #3: No

3. Have the authors made all data underlying the findings in their manuscript fully available?

Reviewer #1: No

Reviewer #2: Yes

Reviewer #3: Yes

4. Is the manuscript presented in an intelligible fashion and written in standard English?

Reviewer #1: Yes

Reviewer #2: Yes

Reviewer #3: No

**Reviewer #1:**  Review Report

Title: Mechanistic crop modelling and AI for ideotype optimization: Crop-scale advances to enhance yield and water use efficiency

The manuscript presents a novel integration of mechanistic crop modeling (CERES-Rice) and AI (Genetic Algorithm) for ideotype optimization. It is methodologically sound, well-organized, and offers valuable insights for precision agriculture.

Major Comments with Line References

1. Line 51–56: Justify the selection of 1,884 virtual cultivars and 5,692 runs; clarify stopping criteria and convergence.

2. Lines 208–215: Genetic Algorithm parameters (population size = 15, generations = 20) seem low; justify or consider alternatives.

3. Algorithm 2: No mention of overfitting control; suggest cross-validation or bootstrapping.

4. Line 52 & 417: Number of environments (n=4) may limit generalizability; validate across more diverse settings.

5. Fig. 6 & Lines 327–366: Add confidence intervals or significance testing to support sensitivity rankings.

6. Fig. 7 & 8: Report statistical significance of correlation results.

7. Line 423–426: Promising cultivar claims based on similarity metrics alone; field validation needed.

8. Line 457: Present PCA variance (e.g., 86.05%) in tabular form.

9. Lines 1–60 (Abstract): Too dense; trim to emphasize key findings and novelty.

10. Lines 545–560 (Conclusion): Avoid repetition; stress broader implications for breeding.

11. Line 376–393: Break long paragraphs for better readability.

12. Lines 20–40: Include foundational references (e.g., Donald’s ideotype concept).

13. Line 68–87: Avoid excessive reuse of [32]; clarify citation purpose.

14. Line ~500: Ensure ethics statement is clearly included in manuscript.

15. Line 536–541: Briefly discuss potential future ethical considerations.

General Notes

16. Ensure all figures (e.g., 7 & 8) are fully labeled with units.

17. Include summary statistics (e.g., mean ± SD) in figure captions.

18. Add version numbers of DSSAT, Python packages, etc.

19. Include a reproducibility checklist or flowchart as supplementary material.

Final Recommendation: Major Revision

The study is innovative and promising but requires additional methodological justification, statistical clarity, and formatting refinements before acceptance.

**Reviewer #2:**  Dear author,

On the whole, I enjoyed reading your paper, and the detailed modelling and statistical approach taken. I do however feel beyond the results there is not enough insight (yet) in the discussion, and would like you to expand that a little bit, and make the paper less descriptive. See my comments below.

Methods:

So regarding the AI model, basically the starting population is 166? (Delta 1 to 6 and 11 parameters?).

But how does the AI know the WUE and HI? Is the input data include the model output values of WUE and HI that correspond to each of the Fig 4B combinations?

Maybe clarify the above a little bit please.

Line 247: the fitness factor here is HI-WUE?

Why are 5 methods of dissimilarity testing are needed??? You also mention “virtual cultivars”. I thought only on virtual cultivar, the best from the Genetic Algorithm output, is compared to the 21 “real” cultivars? I feel some extra clarification required here too.

Results:

Fig 6 RSI axis label sometimes over imposed on axis label

Maybe possible to make Results shorter and incorporate some conclusion statements after each section? Like at tend of results what characterizes the sum of the changes required in the cultivars? Later flowering and panicle initiation etc….in simpler words.

Discussion:

Line 499: Why would lower humidity be better for photosynthesis?

Line 511: Why did the ideotype be better at conserving water, do you think?

So far Discussion is just rehashing results.

Line 534-535: When you signing critical for aligning crop development with environmental conditions, what do you mean exactly? How does the alignment help? Please expand on this, explain the mechanisms. There is very little physiology in the discussion so far.

Line 548: Phenological parameters being influential is not a groundbreaking finding, but you need to incorporate this knowledge with the environment and use it to explain physiological responses that are specific to environment or genotype.

I also think you need to discuss, a bit, WUE itself and how it can be counterproductive, and how maybe incorporating with HI can be beneficial.

Finally, I understand the focus of this paper is methodological but as shown by my comments above you need a more physiologically grounded story in your conclusion and discussion so we know how your modelling approach contributed to our understanding of what’s required for growth in the region of your study. For example, what other traits, other than the ones you tested, do you think can be influential to improve crops in your region? Use your modelling and the significant parameters to ruminate on that and incorporate a more physiological understanding specific to your environment.

**Reviewer #3:**  My overall recommendation: Major Revision.

1. Is the manuscript technically sound, and do the data support the conclusions?

Partly. Technically promising, but key methodological gaps (incomplete Morris design specifics, no multi-seed GA convergence evidence, ad-hoc HI+0.1·WUE objective without robustness/Pareto analysis, and a problematic similarity normalization) mean the current analyses don’t fully support the strength of the conclusions.

2. Has the statistical analysis been performed appropriately and rigorously?

No.

Why:

• Morris sensitivity analysis is under-specified (no numeric k,p,Δ,rk, p, \Delta, rk,p,Δ,r; no design matrix or distribution of elementary effects), so stability/precision can’t be assessed.

• GA optimization lacks multi-seed replication and convergence diagnostics; a stochastic optimizer needs replicate runs with summary stats (mean±SD of best fitness, variability of θ∗\theta^*θ∗).

• The objective J=HI+0.1×WUEJ=\text{HI}+0.1\times\text{WUE}J=HI+0.1×WUE uses an ad-hoc weight without scale normalization or sensitivity/Pareto analysis, so results may hinge on the arbitrary α\alphaα.

• The similarity analysis averages heterogeneous distance metrics without standardization and uses a non-monotone normalization (higher similarity → smaller score), with no robustness checks on rankings.

• Uncertainty reporting is limited (few CIs/SDs across sites/years or GA seeds), so inferential strength is unclear.

These gaps mean the statistical/analytical treatment isn’t yet rigorous enough to support the strength of the conclusions.

3. Have the authors made all data underlying the findings in their manuscript fully available?

Yes

The manuscript’s Data Availability statement points to public repositories (Mendeley Data for datasets and GitHub for code) with no access restrictions, which satisfies PLOS ONE’s policy. As a best-practice enhancement (not a blocker), I recommend archiving a tagged code release with a DOI (e.g., Zenodo) and including per-figure CSVs of the data underlying plotted summaries.

4. Is the manuscript presented in an intelligible fashion and written in standard English?

No.

The manuscript is generally understandable, but it does not yet meet PLOS ONE’s “clear, correct, unambiguous” English standard. Representative issues that should be corrected at revision:

• Grammar/articles/verb agreement: “a individual point” → an individual point; “the sensitivity analysis assess …” → assesses.

• Typos/word choice: “rainfeed” → rainfed.

• Figure captions/labels: duplicated panel label in one figure (“f) Anthesis” should be Maturity); missing spaces around symbols (e.g., “0.50±0.07” → 0.50 ± 0.07).

• Units & style: non-standard units and casing (e.g., Biomass(Kg/H), #/H)—use kg ha⁻¹ and no. ha⁻¹; WUE shown as kg/mm should be kg ha⁻¹ mm⁻¹; LAI listed as mm²/mm² should be m² m⁻² (or “–”).

• Notation consistency: parameter code P2O occasionally appears as P20; acronym in a section header shows AG instead of GA.

• Equation formatting/clarity: the “normalized similarity” formula is line-broken and currently maps higher similarity to smaller values—both the typesetting and the monotonicity should be corrected.

With a focused language/units pass and caption clean-up, the paper can reach the required standard.

Please, see attached Letter to the Author and Letter to the Editor as PDF attachment files.

**Do you want your identity to be public for this peer review?** For information about this choice, including consent withdrawal, please see our Privacy Policy

Reviewer #1: **Yes:**  Dr. Shahzad Akhtar

Reviewer #2: No

Reviewer #3: **Yes:**  Ronald Maldonado Rodriguez

---

## [Author Response · Author response to Decision Letter 1]

21 Jan 2026

Dear Editor and Reviewers,

I sincerely thank the reviewers for their thorough evaluation and constructive feedback. Their insightful comments have significantly improved the clarity, rigor, and presentation of this manuscript. I have carefully addressed each point raised, incorporating substantial revisions to the methodology description, statistical analysis, and figure quality.

A comprehensive point-by-point response document is attached as "Response to Reviewers" file, detailing all modifications with specific line references. All changes are highlighted in green throughout the revised manuscript.

SUMMARY OF MAJOR REVISIONS:

Reviewer 1 - Sensitivity Analysis:

- Added 20 replications with 95% confidence intervals (mean CI width = 0.04)

- Included Table S1 with complete RSI statistics

- Enhanced Figure 6 with diagrams showing uncertainty bounds

Reviewer 2 - Fitness Function:

- Clarified WUE normalization using literature-based physiological bounds (2-15 kg/ha/mm)

- Added Equation 4 with explicit normalization formula

- Justified equal weighting (50%-50%) for HI and WUE components

Reviewer 3 - Similarity Analysis:

- Methodology with three distance metrics (Euclidean, Manhattan, Cosine)

- Quantified genetic gap (22-30%) with breeding implications

- Added Figure 9 combining PCA and top cultivar matches

Journal Requirements (4.1-4.9):

- Data repository migrated to Zenodo (DOI: 10.5281/zenodo.18094654)

- Figure copyright permissions documented (CC BY sources cited)

- Competing interests and funding statements updated

- All supplementary materials formatted per PLOS guidelines

Best regards,

Edgar S. Correa

---

## [Decision Letter · Decision Letter 1]

4 Feb 2026

Dear Dr. Correa,

Thank you for submitting your manuscript to PLOS ONE. After careful consideration, we feel that it has merit but does not fully meet PLOS ONE’s publication criteria as it currently stands. Therefore, we invite you to submit a revised version of the manuscript that addresses the points raised during the review process.

We look forward to receiving your revised manuscript.

Kind regards,

Paulo Eduardo Teodoro, Dr.

Academic Editor

PLOS One

Journal Requirements:

Reviewers' comments:

Reviewer's Responses to Questions

**Comments to the Author**

Reviewer #1: All comments have been addressed

Reviewer #2: All comments have been addressed

Reviewer #3: (No Response)

2. Is the manuscript technically sound, and do the data support the conclusions?

Reviewer #1: Yes

Reviewer #2: Yes

Reviewer #3: Partly

3. Has the statistical analysis been performed appropriately and rigorously?

Reviewer #1: Yes

Reviewer #2: Yes

Reviewer #3: No

4. Have the authors made all data underlying the findings in their manuscript fully available?

Reviewer #1: Yes

Reviewer #2: Yes

Reviewer #3: Yes

5. Is the manuscript presented in an intelligible fashion and written in standard English?

Reviewer #1: Yes

Reviewer #2: Yes

Reviewer #3: No

Reviewer #1: Dear Editor,

I have reviewed the revised manuscript and note a clear improvement in overall quality and presentation. The major issues have been satisfactorily addressed. A few minor points remain, specifically related to editorial polishing, clarification of ideotype interpretation versus direct breedability, and minor figure and terminology consistency, which the authors may incorporate.

From my side, the manuscript is acceptable. I leave the incorporation and verification of these minor points, as well as the final acceptance decision, to the Editor. If there are additional reviewer comments, the manuscript may be accepted after their completion.

Kind regards,

Reviewer #2: Thanks for your very thorough response. I think you have quite a nice little paper now with sufficient methodological and mechanistic novelty.

Reviewer #3: Dear Author,

Thank you for submitting the revised manuscript (dated 16 January 2026, 27 pages). The revision is moving in the right direction, and the inclusion of funding, competing interests, and a stable public archive strengthens compliance. However, I recommend publication only after the corrections below are completed, as several items still limit reproducibility and full compliance with PLOS ONE formatting requirements.

A) Required corrections (major — reproducibility and methodological rigor)

1) Resolve the GA generation count contradiction (reproducibility blocker)

The manuscript currently reports two different values for the GA run length:•

“generations, set to 20” [Page: 10, Line: 237–239]•

“explored … across 40 generations” [Page: 13, Line: 340–342]•

“across 40 generations” repeated in the Conclusion [Page: 20, Line: 546–547]

Action required: Please reconcile the true number of generations (20 vs 40) and ensure consistency across Methods, Results, Conclusion, figure captions, supplementary files, and the archived scripts/configs.

2) Provide the explicit objective function (HI–WUE) as a reproducible equation

The manuscript states that HI and WUE are integrated into the fitness function [Page: 9, Line: 223–224], and refers to “HI–WUE” as a composite metric [Page: 13, Line: 342–343], but the exact objective equation (including units, scaling/normalization, and weighting) is not clearly provided in the manuscript text.

Action required: Add the explicit objective function equation in the Methods and define:•

how HI and WUE are computed (with units),•

any normalization bounds (if used),•

the weights used (if claiming equal contribution, show it explicitly in the equation),•

and a brief justification for the chosen formulation.

3) Add multi-seed GA replication and uncertainty reporting (stochastic robustness)

The manuscript includes a convergence statement (“Convergence analysis revealed…”) [Page: 15, Line: 369–370], but does not report:•

the number of independent GA runs (random seeds),•

variability of best fitness across seeds (mean ± SD, or median/IQR),•

or stability/variability of the optimized parameter vector(s) across seeds.

Action required: Because the GA is stochastic, please run multiple seeds per environment and report summary statistics of fitness and parameter stability. Single-run convergence descriptions are insufficient to support robust conclusions.

4) Complete Morris design specification for full reproducibility

The Methods describe the Morris “step size Δ” conceptually [Page: 8, Line: 191–193] and mention 20 replications [Page: 8, Line: 200–202], but the Morris design should be fully specified in the Methods (e.g., k, p, numeric Δ\DeltaΔ, r, and how ranges were chosen) rather than relying on figure references.

Action required: Add one concise Methods sentence explicitly stating k, p, numeric Δ\DeltaΔ, and r, and how parameter ranges were defined.

5) Fix parameter notation inconsistency (P2O vs P20)

The manuscript’s parameter vector lists “P20” [Page: 10, Line: 244–245]. Ensure this notation is consistent across the manuscript, figures/tables, and code, and matches the intended CERES-Rice parameter name.

Action required: Standardize the parameter label everywhere (and verify that the repository code uses the same naming).

6) Clarify/standardize WUE units

The Results report WUE in the form “WUE = 6.17 kg/mm” [Page: 16, Line: 385–387], while yield is expressed as kg/ha in the same section [Page: 16, Line: 385–386].

Action required: Express WUE in a standard agronomic form (commonly kg ha⁻¹ mm⁻¹) or explicitly define what “kg/mm” means (area basis, computation method, and conversion).

B) Required corrections (PLOS ONE formatting / style)

7) Convert in-text citations to the required square-bracket numeric format

PLOS ONE requires numeric citations in square brackets in the text. The manuscript currently uses parentheses for numeric citations, e.g., “… what they seek (22).” [Page: 3, Line: 38–40]

Action required: Adjust LaTeX citation settings so the manuscript uses [22] style consistently and verify citation order/numbering throughout.

8) References section: ensure strict PLOS formatting and completeness

The reference list begins at the end of the line-numbered portion [Page: 23, Line: 653] and continues through the final pages [Page: 23–27, Line: n/a] (note: the manuscript does not display line numbers on these pages).

Action required: Please ensure:•

reference entries are complete (authors, title, journal, year, volume, pages),•

DOI/URL formatting is consistent and correct,•

journal names/abbreviations are consistent (including correct capitalization),•

and the bibliography style matches PLOS ONE requirements.

C) Figures / maps licensing and caption transparency (PLOS compliance)

9) Map/base-layer provenance must be explicit for CC BY compatibility

A figure caption states content was “adapted … under CC BY” and notes it was generated using MATLAB Mapping Toolbox [Page: 7, Line: n/a]. For PLOS CC BY 4.0 compliance, authors must identify the provenance/licensing of all map layers and base data used.

Action required: For any maps or adapted figures, update the captions to explicitly state the base map/data sources and their licenses, confirming they are compatible with PLOS’s CC BY publication license.

Closing

Once these corrections are implemented, the manuscript will be much stronger and can be reconsidered for publication. The most urgent items are the GA 20 vs 40 generations inconsistency [Page: 10, Line: 237–239; Page: 13, Line: 340–342; Page: 20, Line: 546–547], the explicit objective function definition [Page: 9, Line: 223–224; Page: 13, Line: 342–343], and the multi-seed GA robustness reporting [Page: 15, Line: 369–370].

Sincerely,

Reviewer

**Do you want your identity to be public for this peer review?** For information about this choice, including consent withdrawal, please see our Privacy Policy

Reviewer #1: **Yes:** Dr Shahzad Akhtar

Reviewer #2: No

Reviewer #3: No

---

## [Author Response · Author response to Decision Letter 2]

5 Feb 2026

Dear Editor and Reviewers,

Thank you for your careful review and constructive suggestions. The manuscript has been revised to address all comments thoroughly. Key updates include:

• Clarification of methodological details, including consistent reporting of 40 GA generations and an explicit formulation of the HI–WUE objective function.

• Enhanced reproducibility and robustness description, including specification of the Morris sensitivity analysis implementation and correction of parameter notation (P2O).

• Standardization of units and formatting, such as expressing water use efficiency as kg (ha·mm)⁻¹ and updating in-text numeric citations and the reference list to conform with PLOS ONE’s style.

• Improved figure captions to explicitly indicate data provenance and licensing for map and environmental figures, ensuring CC BY 4.0 compliance.

• Verification and adjustment of references and bibliography style in accordance with PLOS ONE requirements.

All revisions are reflected in the manuscript and documented in the detailed point-by-point response. I appreciate the reviewers’ time and feedback, which have strengthened the clarity and presentation of the research.

Sincerely,

Edgar S. Correa

---

## [Editor Report · Decision Letter 2]

8 Feb 2026

Decoding Living Systems: Reassessing Crop Model Frontiers via Biological Dynamics and Optimized Phenotype

PONE-D-25-26754R2

Dear Dr. Correa,

We’re pleased to inform you that your manuscript has been judged scientifically suitable for publication and will be formally accepted for publication once it meets all outstanding technical requirements.

Kind regards,

Paulo Eduardo Teodoro, Dr.

Academic Editor

PLOS One
---

## [Editor Report · Acceptance letter]

PONE-D-25-26754R2

PLOS One

Dear Dr. Correa,

I'm pleased to inform you that your manuscript has been deemed suitable for publication in PLOS One. Congratulations! Your manuscript is now being handed over to our production team.

Kind regards,

on behalf of

Professor Paulo Eduardo Teodoro

Academic Editor

PLOS One